# Privacy of Noisy Stochastic Gradient Descent: More Iterations without More Privacy Loss

**Jason M. Altschuler**
MIT
jasonalt@mit.edu

**Kunal Talwar**
Apple
ktalwar@apple.com

## Abstract

A central issue in machine learning is how to train models on sensitive user data. Industry has widely adopted a simple algorithm: Stochastic Gradient Descent with noise (a.k.a. Stochastic Gradient Langevin Dynamics). However, foundational theoretical questions about this algorithm's privacy loss remain open—even in the seemingly simple setting of smooth convex losses over a bounded domain. Our main result resolves these questions: for a large range of parameters, we characterize the differential privacy up to a constant. This result reveals that all previous analyses for this setting have the wrong qualitative behavior. Specifically, while previous privacy analyses increase ad infinitum in the number of iterations, we show that after a small burn-in period, running SGD longer leaks no further privacy. Our analysis departs from previous approaches based on fast mixing, instead using techniques based on optimal transport (namely, Privacy Amplification by Iteration) and the Sampled Gaussian Mechanism (namely, Privacy Amplification by Sampling). Our techniques readily extend to other settings.

## 1 Introduction

Convex optimization is a fundamental task in machine learning. When models are learnt on sensitive data, privacy becomes a major concern—motivating a large body of work on differentially private convex optimization [3, 4, 5, 8, 13, 19, 21, 30]. In practice, the most common approach for training private models is NOISY-SGD, i.e., Stochastic Gradient Descent with noise added each iteration. This algorithm is simple and natural, has optimal utility bounds [3, 4], and is implemented in mainstream machine learning platforms such as Tensorflow (TF Privacy) [1], PyTorch (Opacus) [35], and JAX [6].

Yet, despite the simplicity and ubiquity of this NOISY-SGD algorithm, we do not understand basic questions about its *privacy loss*[1]—i.e., how sensitive the output of NOISY-SGD is with respect to the training data. Specifically:

**Question 1.1.** *What is the privacy loss of* NOISY-SGD *as a function of the number of iterations?*

Even in the seemingly simple setting of smooth convex losses over a bounded domain, this fundamental question has remained wide open. In fact, even more basic questions are open:

**Question 1.2.** *Does the privacy loss of* NOISY-SGD *increase ad infinitum in the number of iterations?*

The purpose of this paper is to understand these fundamental theoretical questions. Specifically, we resolve Questions 1.1 and 1.2 by characterizing the privacy loss of NOISY-SGD up to a constant factor in this (and other) settings for a large range of parameters. Below, we first provide context by describing previous analyses in §1.1, and then describe our result in §1.2 and techniques in §1.3.

---

The Supplementary Material contains a full version of this paper with proofs and more discussion.

[1]Throughout, we follow the literature by writing "privacy loss" to refer to the differential privacy parameters.

36th Conference on Neural Information Processing Systems (NeurIPS 2022).

## 1.1 Previous approaches and limitations

Although there is a large body of research devoted to understanding the privacy loss of NOISY-SGD, all existing analyses have (at least) one of the following two drawbacks.

**Privacy bounds that increase ad infinitum.** One type of analysis approach yields upper bounds on the DP (resp., Rényi DP) that scale as $\sqrt{T}$ (resp., $T$). This includes the original analyses of NOISY-SGD, which were based on the techniques of Privacy Amplification by Sampling and Advanced Composition [2, 3], as well as alternative analyses based on the technique of Privacy Amplification by Iteration [12]. A key issue with these analyses is that they increase unboundedly in the number of iterations $T$. This limits the number of iterations that NOISY-SGD can be run given a reasonable privacy budget, typically leading to suboptimal optimization error in practice. Is this a failure of existing analysis techniques or an inherent fact about the privacy loss of NOISY-SGD?

**Privacy bounds that apply for large $T$ and require strong additional assumptions.** The second type of analysis approach yields convergent upper bounds on the privacy loss, but requires strong additional assumptions. This approach is based on connections to sampling. The high-level intuition is that NOISY-SGD is a discretization of a continuous-time algorithm with bounded privacy loss. Specifically, NOISY-SGD can be interpreted as the Stochastic Gradient Langevin Dynamics (SGLD) algorithm [33], which is a discretization of a continuous-time Markov process whose stationary distribution is equivalent to the exponential mechanism [23] and thus is differentially private under certain assumptions.

However, making this connection precise requires strong additional assumptions and/or the resolution of longstanding open questions about the mixing time of SGLD (see §1.4 for details). Only recently did a large effort in this direction culminate in the breakthrough work by Chourasia et al. [9] which proves that *full batch*[2] Langevin dynamics (a.k.a., NOISY-GD rather than NOISY-SGD) has a privacy loss that converges as $T \to \infty$ in this setting where the smooth losses are additionally assumed to be *strongly convex*.

Unfortunately, the assumption of strong convexity seems unavoidable with current techniques. Indeed, in the absence of strong convexity, it is not even known if NOISY-SGD converges to a private stationary distribution, let alone if this convergence occurs in a reasonable amount of time. (The tour-de-force work [7] shows mixing in (large) polynomial time, but only in total variation distance which does not have implications for privacy.) There are fundamental challenges for proving such a result. In short, SGD is only a weakly contractive process without strong convexity, which means that its instability increases with the number of iterations [18]—or in other words, it is plausible that NOISY-SGD could run for a long time while memorizing training data, which would of course mean it is not a privacy-preserving algorithm. As such, given state-of-the-art analyses in both sampling and optimization, it is unclear if the privacy loss of NOISY-SGD should even remain bounded; i.e., it is unclear what answer one should even expect for Question 1.2, let alone Question 1.1.

## 1.2 Contributions

The purpose of this paper is to resolve Questions 1.1 and 1.2. To state our result requires first recalling the parameters of the problem. Throughout, we prefer to state our results in terms of Rényi Differential Privacy (RDP); these RDP bounds are easily translated to DP bounds, as mentioned below in Remark 1.4. See the preliminaries section §2 for definitions and background on privacy.

We consider the basic NOISY-SGD algorithm run on a dataset $\mathcal{X} = \{x_1, \ldots, x_n\}$, where each $x_i$ defines a convex, $L$-Lipschitz, and $M$-smooth loss function $f_i(\cdot)$ on a convex set $\mathcal{K}$ of diameter $D$. For any step size $\eta \leqslant 2/M$, batch size $b$, and initialization $\omega_0 \in \mathcal{K}$, we iterate $T$ times the update

$$\omega_{t+1} \leftarrow \Pi_{\mathcal{K}}[\omega_t - \eta(G_t + Z_t)],$$

where $G_t$ denotes the average gradient vector on a random batch of size $b$, $Z_t \sim \mathcal{N}(0, \sigma^2 I_d)$ is an isotropic Gaussian, and $\Pi_{\mathcal{K}}$ denotes the Euclidean projection onto $\mathcal{K}$.

---

[2]Recently, Ye and Shokri [34] and Ryffel et al. [27], in works concurrent to the present paper, extended the result of [9] to SGLD by removing the full batch assumption; we also obtain the same result by a direct extension of our (completely different) techniques, see the Supplementary Materials. Note that both papers [27, 34] still require strongly convex losses, and in fact state in their conclusions that removing this assumption is an open problem that "would pave the way for wide adoption by data scientists." Our main result resolves this question.

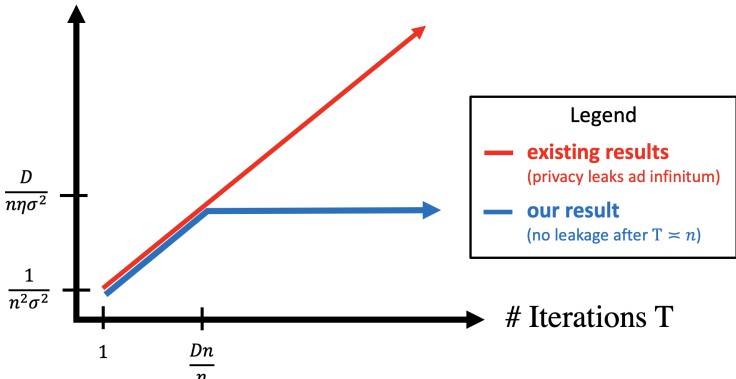

Figure 1: Even in the basic setting of smooth convex optimization over a bounded domain, existing implementations and theoretical analyses of NOISY-SGD—the standard algorithm for private optimization—leak privacy ad infinitum as the number of iterations $T$ increases. Our main result establishes that after a small burn-in period of $\bar{T} = \Theta(n)$ iterations, there is no further privacy loss. For simplicity, this plot sets $L = 1$ (wlog by rescaling), $\alpha = O(1)$ (the regime in practice), and omits logarithmic factors; see the main text for our precise (and optimal) dependence on all parameters.

Known privacy analyses of NOISY-SGD give an $(\alpha, \varepsilon)$-RDP upper bound of

$$\varepsilon \lesssim \frac{\alpha L^2}{n^2 \sigma^2} T \tag{1.1}$$

which increases ad infinitum as the number of iterations $T \to \infty$ [2, 3, 12]. Our main result is a tight characterization of the privacy loss, answering Question 1.1. This result also answers Question 1.2 and shows that previous analyses have the wrong qualitative behavior: after a small burn-in period of $\bar{T} = \Theta\left(\frac{nD}{L\eta}\right)$ iterations, NOISY-SGD leaks no further privacy. See Figure 1.

**Theorem 1.3** (Informal statement of main result: tight characterization of the privacy of NOISY-SGD). *For a large range of parameters,* NOISY-SGD *satisfies* $(\alpha, \varepsilon)$-*RDP for*

$$\varepsilon \lesssim \frac{\alpha L^2}{n^2 \sigma^2} \min\left\{T, \frac{Dn}{L\eta}\right\}, \tag{1.2}$$

*and moreover this bound is tight up to a constant factor.*

Observe that the privacy bound in Theorem 1.3 is identical to the previous bound (1.1) when the number of iterations $T$ is small, but stops increasing when $T \gtrsim \bar{T}$. Intuitively, $\bar{T}$ can be interpreted as the smallest number of iterations required for two NOISY-SGD processes run on adjacent datasets to, with reasonable probability, reach opposite ends of the constraint set.[3] In other words, $\bar{T}$ is effectively the smallest number of iterations before the final iterates could be maximally distinguishable.

To prove Theorem 1.3, we show a matching upper bound (in §3) and lower bound (in §4). These two bounds are formally stated as Theorems 3.1 and 4.1, respectively. See §1.3 for an overview of our techniques and how they depart from previous approaches.

We conclude this section with several remarks about Theorem 1.3.

**Remark 1.4** (Tight DP characterization). *While Theorem 1.3 characterizes the privacy loss of* NOISY-SGD *in terms of RDP, our results can be restated in terms of standard DP bounds. Specifically, by a standard conversion from RDP to DP [25, Proposition 1], if $\delta$ is not smaller than exponentially small in $-b\sigma/n$ and if the resulting bound is $\varepsilon \lesssim 1$, it follows that* NOISY-SGD *is $(\varepsilon, \delta)$-DP for*

$$\varepsilon \lesssim \frac{L}{n\sigma} \sqrt{\min\left\{T, \frac{Dn}{L\eta}\right\} \log 1/\delta}. \tag{1.3}$$

*A matching DP lower bound is proved along the way when we establish our RDP lower bound in §4.*

---

[3]Details: NOISY-SGD updates on adjacent datasets differ by at most $\eta L/b$ if the different datapoint is in the current batch (i.e., with probability $b/n$), and otherwise are identical. Thus, in expectation and with high probability, it takes $\bar{T} \asymp (Dn)/(\eta L)$ iterations for the two NOISY-SGD processes to differ by distance $D$.

**Discussion of assumptions in Theorem 1.3.**

Boundedness. All previous privacy analyses are oblivious to any sort of diameter bound and therefore unavoidably increase ad infinitum[4], c.f. our lower bound in Theorem 1.3. Our analysis is the first to exploit boundedness: whereas previous analyses can only argue that having a smaller constraint set does not worsen the privacy loss, we show that this strictly improves the privacy loss, in fact making it convergent as $T \to \infty$. See the techniques section §1.3. We emphasize that this diameter bound is a mild constraint. Indeed, every utility/optimization guarantee for (non-strongly) convex losses inevitably has a similar dependence, simply due to the difference between initialization and optimum. We also mention that one can solve an unconstrained problem by solving constrained problems with norm bounds, paying only a small logarithmic overhead on the number of solves and a constant overhead in the privacy loss using known techniques [22]. Moreover, in many optimization problems, the solution set is naturally constrained either from the problem formulation or application.

Smoothness. The smoothness assumption on the losses can be relaxed by running NOISY-SGD on a smoothed version of the objective. This can be done using standard techniques, e.g., Gaussian convolution smoothing [12, §5.5], or Moreau-Yousida smoothing by replacing gradient steps with proximal steps [4, §4].

Convexity. Convexity appears to be essential for privacy bounds that do not increase ad infinitum in $T$. Our analysis uses convexity in an essential way to ensure that NOISY-SGD is a contractive process. However, convexity is too restrictive when training deep networks, and it is an interesting open question if the assumption of convexity be relaxed. Any such result appears to require entirely new techniques—if even true. The key technical challenge is that for any iterative process whose set of reachable fixed points is non-convex, there will be non-contractivity at the boundary between basins of attraction—and this precludes arguments based on Privacy Amplification by Iteration.

Lipschitzness. This assumption can be safely removed since smoothness and boundedness imply Lipschitzness for $L = MD$. We write our result in this way in order to clearly isolate where this dependence comes from, and also because the Lipschitz parameter $L$ may be much better than $MD$.

Mild assumptions on parameters. The lower bound makes the mild assumption that the diameter $D$ of the decision set is neither asymptotically smaller than the movement size from one gradient step, nor asymptotically smaller than the standard deviation from $\bar{T}$ random increments of Gaussian noise $\mathcal{N}(0, \eta^2\sigma^2)$ so that the noise does not overwhelm the learning. The upper bound uses the technique of Privacy Amplification by Sampling, and thus inherits an upper bound assumption on $\alpha$ from the analysis of the Sampled Gaussian mechanism, as well as mild bounds on the batch size $b$ that only affect the complete range $\{1, \ldots, n\}$ up to a constant factor assuming that the noise $\sigma$ is not so small that it is asymptotically overwhelmed by one gradient step. These restrictions neither affect numerical bounds which can be computed for any $\alpha$ and $b$ (see §2.3), nor do they affect the asymptotic $(\varepsilon, \delta)$-DP bounds in most parameter regimes of interest.

Extensions. Our analysis techniques extend to related settings, as will be investigated in an extended journal version of this paper. We mention here that if the convexity assumption on the losses is replaced by strong convexity, then NOISY-SGD enjoys better privacy. Specifically, Theorem 1.3 extends identically except that the threshold $\bar{T}$ for no further privacy loss improves from linear to *logarithmic* in $n$, namely $\tilde{O}(\kappa)$ where $\kappa$ denotes the condition number of the losses. This matches the independent results of [27, 34] (see Footnote 2), and uses completely different techniques from them.

### 1.3 Techniques

**Upper bound on privacy.** Our analysis technique departs from recent approaches based on fast mixing. This allows us to bypass the many longstanding technical challenges discussed in §1.1.

Instead, our analysis combines the techniques of Privacy Amplification by Iteration and Privacy Amplification by Sampling. As discussed in §1.1, previous analyses based on these techniques yield loose privacy bounds that increase ad infinitum in $T$. Indeed, bounds based on Privacy Amplification by Sampling inevitably diverge since they "pay" for releasing the entire sequence of $T$ iterates, each of which leaks more information about the private data [2, 3]. Privacy Amplification by Iteration avoids releasing the entire sequence by directly arguing about the final iterate; however, previous

---

[4]For strongly convex losses, boundedness is unnecessary for convergent privacy; see the Supplement.

arguments based on this technique are unable to exploit a diameter bound on the constraint set, and therefore inevitably lead to privacy bounds which grow unboundedly in $T$ [12].

Our analysis is based on the observation that when the iterates are constrained to a bounded domain, we can combine the techniques of Privacy Amplification by Iteration and Privacy Amplification by Sampling in order to only pay for the privacy loss from the final $\bar{T} = (Dn)/(L\eta)$ iterations. Specifically, we partition the execution of NOISY-SGD into the initial $T - \bar{T}$ iterations and the final $\bar{T}$ iterations. Recall that to show privacy, we must show that NOISY-SGD produces a similar output if run on a given dataset or another dataset that is changed slightly. On one hand, if the partially learned model $\omega_{T-\bar{T}}$ were fixed, then the privacy loss from the last $\bar{T}$ iterations could be bounded by the standard analysis of NOISY-SGD based on Privacy Amplification by Sampling and Advanced Composition. On the other hand, irrespective of the input dataset, $\omega_{T-\bar{T}}$ lies in a set of diameter $D$. If the remaining iterations were now run on the same dataset, the Privacy Amplification by Iteration Argument can be used to show that the algorithm "forgets" this initial displacement by up to $D$. (This is formalized in Proposition 3.2, which is the first Privacy Amplification by Iteration argument that is not vacuous as $T \to \infty$). In other words, we can control the privacy loss if we change the dataset either in the first $T - \bar{T}$ iterations or in the final $\bar{T}$ iterations. At a high-level, our analysis proceeds by carefully running these arguments in parallel.

**Lower bound on privacy.** We construct two adjacent datasets on which the NOISY-SGD iterates are generated by random walks—one symmetric, one biased—constrained within an interval of length $D$. In this way, we reduce the question of how large is the privacy loss of NOISY-SGD, to the question of how distinguishable is a constrained symmetric random walk from a constrained biased random walk. The core technical challenge is that the distributions of the iterates of these random walks are intractable to reason about explicitly—due to the highly non-linear interactions between the projections and random increments. Briefly, our key technique here is a way to modify these processes so that on one hand, their distinguishability is essentially the same, and the other hand, no projections occur with high probability—allowing us to explicitly compute their distributions and thus also their distinguishability. Details in §4.

## 1.4 Other related work

**Private sampling.** The mixing time of (stochastic) Langevin Dynamics has been extensively studied in recent years starting with [10, 11], and other more recent works have analyzed mixing in other notions of distance. From the point of view of privacy, more stringent notions of mixing are needed, and mixing in Rènyi divergence was studied by Vempala and Wibisono [32] and Ganesh and Talwar [15]. In addition to the aforementioned works, several recent works have studied algorithms for sampling from the distribution $\exp(\varepsilon \sum_i f_i(\omega))$ or its regularized versions from a privacy viewpoint. [24] proposed a mechanism of this kind that works when $f$ may be unbounded and showed that it satisfies $(\varepsilon, \delta)$-DP. Recently, [17] gave better privacy bounds for such a regularized exponential mechanism, and designed an efficient sampler based only on function evaluation. [16] showed a number of results about the continuous Langevin Diffusion, showing that it gives optimal utility bounds for different variants of private optimization problems with different parameters.

As alluded to in §1.1, there are several core issues with trying to prove DP bounds for NOISY-SGD by directly combining "fast mixing" bounds with "private once mixed" bounds. First, mixing results typically do not apply, e.g., since DP requires mixing in stringent divergences like Rényi, or because realistic settings with constraints, stochasticity, and lack of strong convexity are difficult to analyze—indeed, understanding the mixing time for such settings is a challenging open problem. Second, even when fast mixing bounds do apply, directly combining them with "private once mixed" bounds unavoidably leads to DP bounds that are loose to the point of being essentially useless (e.g., the inevitable dimension dependence in mixing bounds to the stationary distribution of the continuous-time Langevin Diffusion would lead to dimension dependence in DP bounds, which should not occur—as we show). Third, even if a Markov chain were private after mixing, one cannot conclude from this that it is private beforehand—indeed, there are simple Markov chains which are private after mixing, yet are exponentially non-private beforehand [14].

**Utility bounds.** In the field of private optimization, one separately analyzes two properties of algorithms: (i) the privacy loss as a function of the number of iterations, and (ii) the utility (a.k.a., optimization error) as a function of the number of iterations. These two properties can then be

combined to obtain privacy-utility tradeoffs. The purpose of this paper is to completely resolve (i); this result can then be combined with any bound on (ii).

Utility bounds for SGD are well understood [28], and these analyses have enabled understanding utility bounds for NOISY-SGD in empirical [3] and population [4] settings. However, there is a big difference between the minimax-optimal utility bounds in theory versus what is desired in practice. Indeed, while in theory a single pass of NOISY-SGD achieves the minimax-optimal population risk [13], in practice NOISY-SGD benefits from running longer to get more accurate training. In fact, this divergence is even true and well-documented for non-private SGD as well, where one epoch is minimax-optimal in theory, but in practice more epochs help. Said simply, this is because typical problems are not worst-case problems (i.e., minimax-optimal theoretical bounds are typically not representative of practice). For these practical settings, in order to run NOISY-SGD longer, it is essential to have privacy bounds which do not increase ad infinitum. Our paper resolves this.

## 2 Preliminaries

In this section, we recall relevant preliminaries about convex optimization (§2.1), differential privacy (§2.2), and two by-now-standard techniques for analyzing the privacy of optimization algorithms—namely, Privacy Amplification by Sampling (§2.3) and Privacy Amplification by Iteration (§2.4).

**Notation.** We write $\mathbb{P}_X$ to denote the law of a random variable $X$, and $\mathbb{P}_{X|Y=y}$ to denote the law of $X$ given the event $Y = y$. We write $Z_{S:T}$ as shorthand for the vector concatenating $Z_S, \ldots, Z_T$. We write $f_{\#}\mu$ to denote the pushforward of a distribution $\mu$ under a (possibly random) function $f$, i.e., the law of $f(X)$ where $X \sim \mu$. We write $\lambda\mu + (1 - \lambda)\nu$ to denote the mixture distribution that is $\mu$ with probability $\lambda$ and $\nu$ with probability $1 - \lambda$. We write $\mathcal{A}(\mathcal{X})$ to denote the output of an algorithm $\mathcal{A}$ run on input $\mathcal{X}$; this is a probability distribution if $\mathcal{A}$ is a randomized algorithm.

### 2.1 Convex optimization

Throughout, the loss function corresponding to a data point $x_i$ or $x_i'$ is denoted by $f_i$ or $f_i'$, respectively. While the dependence on the data point is arbitrary, the loss functions are assumed to be convex in the argument $\omega \in \mathcal{K}$ (a.k.a., the machine learning model we seek to train). Throughout, the set $\mathcal{K}$ of possible models is a convex subset of $\mathbb{R}^d$. In order to establish both optimization and privacy guarantees, two additional assumptions are required on the loss functions. Recall that a differentiable function $g : \mathcal{K} \to \mathbb{R}^d$ is: $L$-Lipschitz if $\|g(\omega) - g(\omega')\| \leqslant L\|\omega - \omega'\|$ for all $\omega, \omega' \in \mathcal{K}$; and $M$-smooth if $\|\nabla g(\omega) - \nabla g(\omega')\| \leqslant M\|\omega - \omega'\|$ for all $\omega, \omega' \in \mathcal{K}$.

### 2.2 Differential privacy and Rényi differential privacy

In order to prove DP guarantees, we work with the related notion of Rényi Differential Privacy (RDP) introduced by [25], since RDP is more amenable to our analysis techniques and is readily converted to DP guarantees [25, Proposition 1]. RDP measures how distinguishable the output of an algorithm is when run on two "adjacent" datasets—i.e., two datasets which differ on at most one datapoint—in Rényi divergence. We therefore first recall the definition of Rényi divergence.

**Definition 2.1** (Rényi divergence)**.** *The Rényi divergence between probability measures $\mu$ and $\nu$ of order $\alpha \in (1, \infty)$ is $\mathcal{D}_\alpha(\mu \parallel \nu) = \frac{1}{\alpha-1} \log \int (\mu(x)/\nu(x))^\alpha \nu(x)dx$, if $\mu \ll \nu$, and $\infty$ otherwise. Here $0/0 = 0$ and $x/0 = \infty$ for $x > 0$. The Rényi divergences of order $\alpha \in \{1, \infty\}$ are defined by continuity.*

**Definition 2.2** (Rényi Differential Privacy)**.** *A randomized algorithm $\mathcal{A}$ satisfies $(\alpha, \varepsilon)$-RDP if $\mathcal{D}_\alpha(\mathcal{A}(\mathcal{X}) \parallel \mathcal{A}(\mathcal{X}')) \leqslant \varepsilon$ for any two adjacent datasets $\mathcal{X}, \mathcal{X}'$.*

Following, we recall two basic properties of RDP that we use in our analysis. Proofs can be found, e.g., in [31, Theorem 9] and [25, Proposition 3], respectively.

**Lemma 2.3** (Post-processing property of Rényi divergence)**.** *For any Rényi parameter $\alpha \geqslant 1$, any (possibly random) function $h$, and any probability distributions $\mu, \nu$,*

$$\mathcal{D}_\alpha(h_{\#}\mu \parallel h_{\#}\nu) \leqslant \mathcal{D}_\alpha(\mu \parallel \nu).$$

**Lemma 2.4** (Strong composition for RDP). *For any Rényi parameter $\alpha \geqslant 1$, and any two sequences of random variables $X_1, \ldots, X_k$ and $Y_1, \ldots, Y_k$,*

$$\mathcal{D}_\alpha \left( \mathbb{P}_{X_{1:k}} \,\middle\|\, \mathbb{P}_{Y_{1:k}} \right) \leqslant \sum_{i=1}^k \sup_{x_{1:i-1}} \mathcal{D}_\alpha \left( \mathbb{P}_{X_i | X_{1:i-1} = x_{1:i-1}} \,\middle\|\, \mathbb{P}_{Y_i | Y_{1:i-1} = x_{1:i-1}} \right).$$

## 2.3 Privacy Amplification by Sampling

A core technique in the DP literature is Privacy Amplification by Sampling [20], which quantifies the idea that a private algorithm run on a small random sample of the input become more private. We recall a convenient statement of this from [26, Theorem 11] in terms of the RDP of the "Sampled Gaussian Mechanism" (which is a composition of subsampling and additive Gaussian noise).

**Definition 2.5** (Rényi Divergence of the Sampled Gaussian Mechanism). *For Rényi parameter $\alpha \geqslant 1$, mixing probability $q \in (0,1)$, and noise parameter $\sigma > 0$, define*

$$S_\alpha(q, \sigma) := \mathcal{D}_\alpha \left( \mathcal{N}(0, \sigma^2) \,\middle\|\, (1-q)\mathcal{N}(0, \sigma^2) + q\mathcal{N}(1, \sigma^2) \right).$$

**Lemma 2.6** (Bound on Rényi Divergence of the Sampled Gaussian Mechanism). *Consider Rényi parameter $\alpha > 1$, mixing probability $q \in (0, 1/5)$, and noise level $\sigma \geqslant 4$. If $\alpha \leqslant \alpha^*(q, \sigma)$, then*

$$S_\alpha(q, \sigma) \leqslant 2\alpha q^2 / \sigma^2.$$

This bound restricts to Rényi parameter at most $\alpha^*(q, \sigma)$, which is defined to be the largest $\alpha$ satisfying $\alpha \leqslant M\sigma^2/2 - \log(\sigma^2)$ and $\alpha \leqslant (M^2\sigma^2/2 - \log(5\sigma^2))/(M + \log(q\alpha) + 1/(2\sigma^2))$, where $M := \log(1 + 1/(q(\alpha-1)))$. While we use Lemma 2.6 to prove the asymptotic bounds in Theorem 1.3, we emphasize that i) our bounds can be computed numerically for *any* $\alpha \geqslant 1$, and ii) this upper bound does not preclude $\alpha$ from the typical parameter regime of interest, see the discussion in §1.2.

## 2.4 Privacy Amplification by Iteration

The Privacy Amplification by Iteration technique of [12] bounds the privacy loss of an iterative algorithm without "releasing" the entire sequence of iterates—unlike arguments based on Privacy Amplification by Sampling, c.f., §1.3. This technique applies to processes generated by a Contractive Noisy Iteration (CNI). We begin by recalling this definition, albeit in a slightly more general form that allows for two differences. The first difference is allowing the contractions to be random; albeit simple, this generalization is critical for analyzing NOISY-SGD because a stochastic gradient update is random. The second difference is that we project each iterate; this generalization is solely for convenience as it simplifies the exposition.[5]

**Definition 2.7** (Contractive Noisy Iteration). *Consider a (random) initial state $X_0 \in \mathbb{R}^d$, a sequence of (random) contractions $\phi_t : \mathbb{R}^d \to \mathbb{R}^d$, a sequence of noise distributions $\xi_t$, and a convex set $\mathcal{K}$. The Projected Contractive Noisy Iteration $\mathrm{CNI}(X_0, \{\phi_t\}, \{\xi_t\}, \mathcal{K})$ is the final iterate $X_T$ of the process*

$$X_{t+1} := \phi_{t+1}(X_t) + Z_{t+1},$$

*where $Z_{t+1}$ is drawn independently from $\xi_{t+1}$.*

**Proposition 2.8** (Original[6] PABI bound). *Let $X_T$ and $X_T'$ denote the outputs of $\mathrm{CNI}(X_0, \{\phi_t\}, \{\xi_t\}, \mathcal{K})$ and $\mathrm{CNI}(X_0, \{\phi_t'\}, \{\xi_t\}, \mathcal{K})$ where $\xi_t = \mathcal{N}(0, \sigma_t^2 I_d)$. Let $s_t := \|\phi_t - \phi_t'\|_\infty$, and consider any sequence $a_1, \ldots, a_T$ such that $z_t := \sum_{i=1}^t (s_i - a_i)$ is non-negative for all $t$ and satisfies $z_T = 0$. Then*

$$\mathcal{D}_\alpha \left( \mathbb{P}_{X_T} \,\middle\|\, \mathbb{P}_{X_T'} \right) \leqslant \frac{\alpha}{2} \sum_{t=1}^T \frac{a_t^2}{\sigma_t^2}.$$

---

[5]Since projections are contractive, these processes could be dubbed Contractive Noisy Contractive Processes (CNCI); however, we use Definition 2.7 as it more closely mirrors previous usages of CNI. Alternatively, since the composition of contractions is a contraction, the projection can be combined with $\phi_t$ to obtain a bona fide CNI; however, this requires defining auxiliary shifted processes which leads to a more complicated analysis.

[6]Strictly speaking, Proposition 2.8 is a generalization of [12, Theorem 22] since it allows for randomized contractions and projections in the CNI. See the Supplementary Materials for a proof.

# 3 Upper bound on privacy

In this section, we prove the upper bound in Theorem 1.3. The formal statement of this result is as follows; see §1.2 for a discussion of the mild assumptions on the parameters.

**Theorem 3.1** (Privacy upper bound for NOISY-SGD). *Let $\mathcal{K} \subseteq \mathbb{R}^d$ be a convex set of diameter $D$, and consider optimizing convex losses over $\mathcal{K}$ that are $L$-Lipschitz and $M$-smooth. Consider any number of iterations $T$, dataset size $n \in \mathbb{N}$, batch size $b \leqslant n$, stepsize $\eta \leqslant 2/M$, noise parameter $\sigma > 0$, and initialization $\omega_0 \in \mathcal{K}$ such that $b < n/5$ and $b\sigma > 8\sqrt{2}L$. Then NOISY-SGD satisfies $(\alpha, \varepsilon)$-RDP if $1 < \alpha \leqslant \alpha^*(\frac{b}{n}, \frac{b\sigma}{2\sqrt{2}L})$ and*

$$\varepsilon \lesssim \frac{\alpha L^2}{n^2 \sigma^2} \min \left\{ T, \frac{Dn}{L\eta} \right\}.$$

## 3.1 Privacy Amplification by Iteration bounds that are not vacuous as $T \to \infty$

Recall from the preliminaries section §2.4 that PABI arguments, while tight for a small number of iterations $T$, provide vacuous bounds as $T \to \infty$ (c.f., Proposition 2.8). The following proposition overcomes this by establishing privacy bounds which are *independent* of the number of iterations $T$. This result only requires additionally making the mild assumption that the CNI iterates are bounded.

**Proposition 3.2** (New PABI bound that is not vacuous as $T \to \infty$). *Let $X_T$, $X'_T$, and $s_t$ be as in Proposition 2.8. Consider any $\tau \in \{0, \ldots, T-1\}$ and sequence $a_{\tau+1}, \ldots, a_T$ such that $z_t := D + \sum_{i=\tau+1}^{t}(s_i - a_i)$ is non-negative for all $t$ and satisfies $z_T = 0$. If $\mathcal{K}$ has diameter $D$, then:*

$$\mathcal{D}_\alpha \left( \mathbb{P}_{X_T} \parallel \mathbb{P}_{X'_T} \right) \leqslant \frac{\alpha}{2} \sum_{t=\tau+1}^{T} \frac{a_t^2}{\sigma_t^2}.$$

**Proof idea.** We first recall the basic idea of the proof of the original PABI bound (Proposition 2.8). Briefly, that proof uses as a potential a certain *shifted Rényi divergence*, which allows for uncertainty in one of the two distributions as measured in optimal transport distance (making it geometrically aware). The original argument bounds the shifted divergence at iteration $T$ (which is precisely the RDP in question), by the shifted divergence at iteration $T - 1$, and so on to the shifted divergence at iteration 0 (which vanishes since NOISY-SGD always has the same initialization). In this way, by keeping track of the privacy loss at each iteration, PABI bounds the RDP. The main idea behind Proposition 3.2 is to change this argument by simply stopping the induction earlier. Specifically, unroll to iteration $\tau$, and then use boundedness of the iterates to control the shifted divergence at that intermediate time $\tau$. For brevity, the proof is deferred to the Supplementary Materials.

We remark that this new version of PABI uses the shift in the shifted Rényi divergence for a different purpose than previous work: rather than just using the shift to bound the bias incurred from updating on two different losses, here we also use the shift to exploit the boundedness of the constraint set.

## 3.2 Proof sketch of Theorem 3.1 (full details in the Supplementary Materials)

**Step 1: Coupling the iterates.** Suppose $\mathcal{X} = \{x_1, \ldots, x_n\}$ and $\mathcal{X}' = \{x'_1, \ldots, x'_n\}$ are adjacent datasets; that is, they agree $x_i = x'_i$ on all indices $i \in [n] \smallsetminus \{i^*\}$ except for at most one index $i^* \in [n]$. Denote the corresponding loss functions by $f_i$ and $f'_i$, where $f_i = f'_i$ except possibly $f_{i^*} \neq f'_{i^*}$. Consider running NOISY-SGD on either $\mathcal{X}$ or $\mathcal{X}'$ for $T$ iterations—call the resulting iterates $\{W_t\}_{t=0}^{T}$ and $\{W'_t\}_{t=0}^{T}$, respectively—where we start from the same point $w_0 \in \mathcal{K}$ and couple the sequence of random batches $\{B_t\}_{t=0}^{T-1}$ and the random noise in each iteration. That is,

$$W_{t+1} = \Pi_{\mathcal{K}}\big[W_t - \tfrac{\eta}{b} \sum_{i \in B_t} \nabla f_i(W_t) + Y_t + Z_t\big]$$

$$W'_{t+1} = \Pi_{\mathcal{K}}\big[W'_t - \tfrac{\eta}{b} \sum_{i \in B_t} \nabla f_i(W'_t) + Y_t + Z'_t\big]$$

for all $t \leqslant T - 1$, where we have split the noise into terms $Y_t \sim \mathcal{N}(0, \eta^2\sigma_1^2)$, $Z_t \sim \mathcal{N}(0, \eta^2\sigma_2^2)$, $Z'_t \sim \mathcal{N}(0, \eta^2\sigma_2^2) + \frac{\eta}{b}\left[\nabla f_{i^*}(W'_t) - \nabla f'_{i^*}(W'_t)\right] \cdot \mathbb{1}_{i^* \in B_t}$, for any $\sigma_1, \sigma_2 > 0$ satisfying $\sigma_1^2 + \sigma_2^2 = \sigma^2$ (we set $\sigma_1 = \sigma_2 = \sigma/\sqrt{2}$ later for asymptotics). In words, this noise-splitting lets us use the noise for both the Privacy Amplification by Sampling and Privacy Amplification by Iteration arguments below.

Importantly, notice that in the definition of $W'_{t+1}$, the gradient is taken w.r.t. loss functions corresponding to data set $\mathcal{X}$ rather than $\mathcal{X}'$; this is corrected via the bias in the noise term $Z'_t$. Notice also that this bias term is only realized (i.e., $Z'_t$ is possibly non-centered) with probability $1 - b/n$ because the probability that $i^*$ is in a random size-$b$ subset of $[n]$ is $b/n$.

**Step 2: Interpretation as conditional CNI sequences.** Observe that conditional on the event that $Z_t = Z'_t$ are equal (call their value $z_t$), then

$$W_{t+1} = \Pi_{\mathcal{K}} \left[ \phi_t(W_t) + Y_t \right]$$
$$W'_{t+1} = \Pi_{\mathcal{K}} \left[ \phi_t(W'_t) + Y_t \right]$$

where

$$\phi_t(\omega) := \omega - \frac{\eta}{b} \sum_{i \in B_t} \nabla f_i(\omega) + z_t. \tag{3.1}$$

This gives us a CNI since $\phi_t$ is contractive. (This contractivity follows from the fact that a stochastic gradient update on a smooth convex function is contractive; details in the Supplementary Materials.)

**Step 3: Bounding the privacy loss.** Recall that we seek to upper bound $\mathcal{D}_\alpha(\mathbb{P}_{W_T} \parallel \mathbb{P}_{W'_T})$. Let $\tau \in \{0, \ldots, T-1\}$ be a threshold parameter to be chosen shortly. Then

$$\mathcal{D}_\alpha \left( \mathbb{P}_{W_T} \parallel \mathbb{P}_{W'_T} \right) \leqslant \mathcal{D}_\alpha \left( \mathbb{P}_{W_T, Z_{\tau:T-1}} \parallel \mathbb{P}_{W'_T, Z'_{\tau:T-1}} \right)$$
$$\leqslant \underbrace{\mathcal{D}_\alpha \left( \mathbb{P}_{Z_{\tau:T-1}} \parallel \mathbb{P}_{Z'_{\tau:T-1}} \right)}_{\textcircled{1}} + \underbrace{\sup_z \mathcal{D}_\alpha \left( \mathbb{P}_{W_T | Z_{\tau:T-1} = z} \parallel \mathbb{P}_{W'_T | Z'_{\tau:T-1} = z} \right)}_{\textcircled{2}} \tag{3.2}$$

Above, the first step is by the post-processing inequality for the Rényi divergence (Lemma 2.3), and the second step is by the strong composition rule for the Rényi divergence (Lemma 2.4).

Step 3a: Bounding the first term in (3.2), using Privacy Amplification by Sampling. In words, this is the privacy loss from releasing $T - \tau$ noisy gradients. This can be bounded by Privacy Amplification by Sampling (Lemma 2.6) and Advanced Composition (Lemma 2.4) via a standard argument [2]. This yields the bound $(T - \tau) S_\alpha \left( \frac{b}{n}, \frac{b\sigma_2}{2L} \right)$; full details in the Supplementary Materials.

Step 3b: Bounding the second term in (3.2), using Privacy Amplification by Iteration. From step 2, we know that conditional on the event that $Z_t = Z'_t$ for all $t \geqslant \tau$, then $\{W_t\}_{t \geqslant \tau}$ and $\{W'_t\}_{t \geqslant \tau}$ are projected CNI (c.f., Definition 2.7) with respect to the same update functions. Thus we may apply the new Privacy Amplification by Iteration bound (Proposition 3.2) with $s_t \equiv 0$ and $a_t \equiv D/(T - \tau)$ to obtain the bound $\alpha D^2/(2\eta^2 \sigma_1^2 (T - \tau))$.

**Step 4: Putting the bounds together.** By plugging into (3.2) the bounds in Steps 3a-3b, we conclude that NOISY-SGD is $(\alpha, \varepsilon)$-RDP for

$$\varepsilon \leqslant \min_{\tau \in \{0, \ldots, T-1\}} \left\{ (T - \tau) S_\alpha \left( \frac{b}{n}, \frac{b\sigma_2}{2L} \right) + \frac{\alpha D^2}{2\eta^2 \sigma_1^2 (T - \tau)} \right\} \tag{3.3}$$

By bounding $S_\alpha \left( \frac{b}{n}, \frac{b\sigma_2}{2L} \right)$ using Lemma 2.6, setting $\sigma_1 = \sigma_2 = \frac{\sigma}{\sqrt{2}}$, setting $\tau = T - \Theta\left( \frac{Dn}{L\eta} \right)$ if $\frac{D}{\eta L} \lesssim \frac{T}{n}$, and otherwise using the simple bound (1.1) which scales linearly in $T$, and simplifying, we obtain the desired privacy bound on $\varepsilon$, proving Theorem 3.1; full details in the Supplementary Materials.

## 4   Lower bound on privacy

In this section, we prove the lower bound in Theorem 1.3. This is stated formally below and holds even for linear loss functions in one dimension, in fact even when all but one of the loss functions are zero. See §1.2 for a discussion of the mild assumptions on the diameter. Below let $\bar{T} := 0.75 \frac{Dn}{L\eta}$.

**Theorem 4.1** (Privacy lower bound for NOISY-SGD). *There exist universal constants $c_D$, $c_\sigma$, $c_\alpha$, $\bar{\alpha}$ and a family of $L$-Lipschitz linear functions over the interval $\mathcal{K} = [-D/2, D/2]$ such that the following holds. Consider running NOISY-SGD from arbitrary initialization $\omega_0$ with any parameters satisfying $D \geqslant c_D \eta L$ and $\sigma^2 \leqslant c_\sigma D^2/(\eta^2 \bar{T})$. Then NOISY-SGD is not $(\bar{\alpha}, \varepsilon)$-RDP for*

$$\varepsilon \leqslant c_\alpha \frac{\bar{\alpha} L^2}{n^2 \sigma^2} \min \left\{ T, \bar{T} \right\}. \tag{4.1}$$

**Construction.** Consider datasets $\mathcal{X} = \{x_1, \ldots, x_{n-1}, x_n\}$ and $\mathcal{X}' = \{x_1, \ldots, x_{n-1}, x_n'\}$ which differ only on $x_n'$, and corresponding functions which are all zero $f_1(\cdot) = \cdots = f_n(\cdot) = 0$, except for $f_n'(\omega) = L(D - \omega)$. Clearly these functions are linear and $L$-Lipschitz. The intuition behind this construction is that running NOISY-SGD on $\mathcal{X}$ or $\mathcal{X}'$ generates a random walk that is clamped to stay within the interval $\mathcal{K}$—but with the key difference that running NOISY-SGD on dataset $\mathcal{X}$ generates a *symmetric* random walk $\{\omega_t\}$, whereas running NOISY-SGD on dataset $\mathcal{X}'$ generates a *biased* random walk $\{\omega_t'\}$ that biases right with probability $b/n$ each step. That is,

$$\omega_{t+1} = \Pi_{\mathcal{K}}\Big[\omega_t + Z_t\Big] \qquad \text{and} \qquad \omega_{t+1}' = \Pi_{\mathcal{K}}\Big[\omega_t' + Y_t + Z_t\Big]$$

where the processes are initialized at $\omega_0 = \omega_0' = 0$, each random increment $Z_t \sim \mathcal{N}(0, \eta^2\sigma^2)$ is an independent Gaussian, and each bias $Y_t$ is $\eta L/b$ with probability $b/n$ and otherwise is 0.

**Key obstacle for analysis: intractability of reasoning about the iterates' distributions.** The high-level intuition behind this construction is simple to state: the bias of the random walk $\{\omega_t'\}$ makes it distinguishable (to the minimax optimal extent, as we show) from the symmetric random walk $\{\omega_t\}$. However, making this intuition precise is challenging because the distributions of the iterates $\omega_t, \omega_t'$ are intractable to reason about explicitly—due to the highly non-linear interactions between the projections and the random increments. Thus we must establish the distinguishability of $\omega_T, \omega_T'$ in a way that avoids reasoning explicitly about their distributions. We show how to do this in the Supplementary Materials and give a full proof of Theorem 4.1 there.

## 5 Discussion

The results of this paper suggest several natural directions for future work:

Clipped gradients? NOISY-SGD implementations sometimes clip gradients to ensure small norms [2]. For generalized linear models, the clipped gradients are gradients of an auxiliary convex loss [29], so our results can be applied directly. However, in general, clipped gradients do not correspond to gradients of a convex loss, in which case our results (as well as all other works in the literature that aim at proving convergent privacy bounds) do not apply. Can this be remedied?

Average iterate? Can similar privacy guarantees be established for the average iterate rather than the last iterate? There are fundamental difficulties with trying to proving this: indeed, the average iterate is provably not as private for NOISY-CSGD [4].

Adaptive stepsizes? Can similar privacy guarantees be established for optimization algorithms with adaptive stepsizes? The main technical obstacle is how to control the privacy loss from how past iterates affect the adaptivity in later iterates. This appears to preclude using our analysis techniques, at least in their current form.

Beyond convexity? Convergent privacy bounds break down without convexity. This precludes applicability to deep neural networks. Is there any hope of establishing similar results under some sort of mild non-convexity? Due to simple non-convex counterexamples where the privacy of NOISY-SGD diverges, any such extension would have to make additional structural assumptions on the non-convexity (and also possibly change the NOISY-SGD algorithm), although it is unclear how this would even look. Moreover, this appears to require significant new machinery as our techniques are the only known way to solve the convex problem, and they break down in the non-convex setting (see also the discussion in §1.2).

General techniques? Can the analysis techniques developed in this paper be used in other settings? Our techniques readily generalize to any iterative algorithm which interleaves contractive steps and noise convolutions. Such algorithms are common in differentially private optimization, and it would be interesting to apply them to other variants of NOISY-SGD.

## Acknowledgments and Disclosure of Funding

We are grateful to Hristo Paskov for many insightful conversations. This work was done in part while JA was an intern at Apple during Summer 2021. JA was also supported in part by NSF Graduate Research Fellowship 1122374, a TwoSigma PhD Fellowship, and an NYU Faculty Fellowship.

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
