} \asymp (Dn)/(L\eta)$ iterations. To explain this, first recall that by definition, differential privacy of NOISY-SGD means that the final iterate of NOISY-SGD is similar when run on two (adjacent but) different datasets from the same initialization. At an intuitive level, we establish this by arguing about the privacy of the following three scenarios:

(i) Run NOISY-SGD for $T$ iterations on different datasets from same initialization.

(ii) Run NOISY-SGD for $\bar{T}$ iterations on different datasets from different initializations.

(iii) Run NOISY-SGD for $\bar{T}$ iterations on same datasets from different initializations.

In order to analyze (i), which as mentioned is the definition of NOISY-SGD being differentially private, we argue that no matter how different the two input datasets are, the two NOISY-SGD iterates at iteration $T - \bar{T}$ are within distance $D$ of each other since they lie within the constraint set $\mathcal{K}$ of diameter $D$. Thus in particular, the privacy of scenario (i) is at most the privacy of scenario (ii), which has initializations that can be arbitrarily different so long as they are within distance $D$ from each other. In this way, we arrive at a scenario which is independent of $T$. This is crucial for obtaining privacy bounds which do not diverge in $T$; however, previously privacy analyses could not proceed in this way because they could not argue about different initialization.

In order to analyze (ii), we use the technique of Privacy Amplification by Sampling—but only on these final $\bar{T} \ll T$ iterations. In words, this enables us to argue that the noisy gradients that NOISY-SGD uses in the last $\bar{T}$ iterations are indistinguishable up to a certain privacy loss (to be precise the RDP scales linearly in $\bar{T}$) despite the fact that they are computed on different input datasets. This enables us to reduce analyzing the privacy of scenario (ii) to scenario (iii).

In order to analyze (iii), we use a new diameter-aware version of Privacy Amplification by Iteration (Proposition 3.2). In words, this shows that running the *same* NOISY-SGD updates on two different initializations masks their initial difference due to both the noise and contraction maps that define each NOISY-SGD update. Of course, this indistinguishability improves the longer that NOISY-SGD is run (to be precise, we show that the RDP scales inversely in $\bar{T}$).[7]

We arrive, therefore, at a privacy loss which is the sum of the bounds from the Privacy Amplification by Sampling argument in step (ii) and the Privacy Amplification by Iteration argument in step (iii). This privacy loss has a natural tradeoff in the parameter $\bar{T}$: the former bound leads to a privacy loss that increases in $\bar{T}$ (since it pays for the release of $\bar{T}$ noisy gradients), whereas the latter bound leads to a privacy loss that decreases in $\bar{T}$ (since more iterations of the same NOISY-SGD process enable better masking of different initializations). Balancing these two privacy bounds leads to the final choice $\bar{T} \asymp \frac{Dn}{L\eta}$. (We remark that this quantity is not just an artefact of our analysis, but in fact is the true answer for when the privacy loss stops increasing, as established by our matching lower bound.)

---

[7]This step of the analysis has a natural interpretation in terms of understanding the mixing time of the Langevin Algorithm; this connection and its implications are explored in detail in [3].

At a high-level, our analysis proceeds by carefully running these arguments in parallel.

We emphasize that this analysis is the first to show that the privacy loss of NOISY-SGD can strictly improve if the constraint set is made smaller. In contrast, previous analyses only argue that restricting the constraint set *cannot worsen* the privacy loss (e.g., by using a post-processing inequality to analyze the projection step). The key technical challenge in exploiting a diameter bound is dealing with the complicated non-linearities that arise when interleaving projections with noisy gradient updates. Our techniques enable such an analysis.

**Lower bound on privacy.**  We construct two adjacent datasets for which the corresponding NOISY-SGD processes are random walks—one symmetric, one biased—that are confined to an interval of length $D$. In this way, we reduce the question of how large is the privacy loss of NOISY-SGD, to the question of how distinguishable is a constrained symmetric random walk from a constrained biased random walk. The core technical challenge is that the distributions of the iterates of these random walks are intractable to reason about explicitly—due to the highly non-linear interactions between the projections and random increments. Briefly, our key technique here is a way to modify these processes so that on one hand, their distinguishability is essentially the same, and the other hand, no projections occur with high probability—allowing us to explicitly compute their distributions and thus also their distinguishability. Details in §4.

## 1.4  Other related work

**Private sampling.**  The mixing time of (stochastic) Langevin Dynamics has been extensively studied in recent years starting with [12, 14]. A recent focus in this vein is analyzing mixing in more stringent notions of distance, such as the Rényi divergence [21, 42], in part because this is necessary for proving privacy bounds. In addition to the aforementioned results, several other works focus on sampling from the distribution $\exp(\varepsilon \sum_i f_i(\omega)$ or its regularized versions from a privacy viewpoint. [30] proposed a mechanism of this kind that works for unbounded $f$ and showed $(\varepsilon, \delta)$-DP. Recently, [23] gave better privacy bounds for such a regularized exponential mechanism, and designed an efficient sampler based only on function evaluation. Also, [22] showed that the continuous Langevin Diffusion has optimal utility bounds for various private optimization problems.

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

An important implication of smoothness is the following well-known fact; for a proof, see e.g., [34].

**Lemma 2.1** (Small gradient steps on smooth convex losses are contractions). *Suppose $f : \mathbb{R}^d \to \mathbb{R}$ is a convex, $M$-smooth function. For any $\eta \leqslant 2/M$, the mapping $\omega \mapsto \omega - \eta\nabla f(\omega)$ is a contraction.*

### 2.2 Differential privacy and Rényi differential privacy

Over the past two decades, differential privacy (DP) has become a standard approach for quantifying how much sensitive information an algorithm leaks about the dataset it is run upon. Differential privacy was first proposed in [16], and is now widely used in industry (e.g., at Apple [35], Google [17], Microsoft [13], and LinkedIn [36]), as well as in government data collection such as the 2020 US Census [43]. In words, DP measures how distinguishable the output of an algorithm is when run on two "adjacent" datasets—i.e., two datasets which differ on at most one datapoint.

**Definition 2.2** (Differential Privacy). *A randomized algorithm $\mathcal{A}$ satisfies $(\varepsilon, \delta)$-DP if for any two adjacent datasets $\mathcal{X}, \mathcal{X}'$ and any measurable event $S$,*

$$\mathbb{P}[\mathcal{A}(\mathcal{X}) \in S] \leqslant e^\varepsilon \mathbb{P}[\mathcal{A}(\mathcal{X}') \in S] + \delta.$$

In order to prove DP guarantees, we work with the related notion of Rényi Differential Privacy (RDP) introduced by [31], since RDP is more amenable to our analysis techniques and is readily converted to DP guarantees. To define RDP, we first recall the definition of Rényi divergence.

**Definition 2.3** (Rényi divergence). *The Rényi divergence between two probability measures $\mu$ and $\nu$ of order $\alpha \in (1, \infty)$ is*

$$\mathcal{D}_\alpha \left( \mu \parallel \nu \right) = \frac{1}{\alpha - 1} \log \int \left( \frac{\mu(x)}{\nu(x)} \right)^\alpha \nu(x) dx,$$

*if $\mu \ll \nu$, and is $\infty$ otherwise. Here we adopt the standard convention that $0/0 = 0$ and $x/0 = \infty$ for $x > 0$. The Rényi divergences of order $\alpha \in \{1, \infty\}$ are defined by continuity.*

**Definition 2.4** (Rényi Differential Privacy). *A randomized algorithm $\mathcal{A}$ satisfies $(\alpha, \varepsilon)$-RDP if for any two adjacent datasets $\mathcal{X}, \mathcal{X}'$,*

$$\mathcal{D}_\alpha \left( \mathcal{A}(\mathcal{X}) \parallel \mathcal{A}(\mathcal{X}') \right) \leqslant \varepsilon.$$

It is straightforward to convert an RDP bound to a DP bound as follows [31, Proposition 1].

**Lemma 2.5** (RDP-to-DP conversion). *Suppose that an algorithm satisfies $(\alpha, \varepsilon_\alpha)$-RDP for some $\alpha > 1$. Then the algorithm satisfies $(\varepsilon_\delta, \delta)$-DP for any $\delta \in (0, 1)$ and $\varepsilon_\delta = \varepsilon_\alpha + (\log \frac{1}{\delta})/(\alpha - 1)$.*

Following, we recall three basic properties of RDP that we use repeatedly in our analysis. The first property regards convexity. While the KL divergence (the case $\alpha = 1$) is jointly convex in its two arguments, for $\alpha \neq 1$ the Rényi divergence is only jointly *quasi*-convex. See e.g., [41, Theorem 13] for a proof and a further discussion of partial convexity properties of the Rényi divergence.

**Lemma 2.6** (Joint quasi-convexity of Rényi divergence). *For any Rényi parameter $\alpha \geqslant 1$, any mixing probability $\lambda \in [0, 1]$, and any two pairs of probability distributions $\mu, \nu$ and $\mu', \nu'$,*

$$\mathcal{D}_\alpha \left( \lambda\mu + (1 - \lambda)\mu' \parallel \lambda\nu + (1 - \lambda)\nu' \right) \leqslant \max \left\{ \mathcal{D}_\alpha \left( \mu \parallel \nu \right), \mathcal{D}_\alpha \left( \mu' \parallel \nu' \right) \right\}.$$

The second property states that pushing two measures forward through the same (possibly random) function cannot increase the Rényi divergence. This is the direct analog of the classic "data-processing inequality" for the KL divergence. See e.g., [41, Theorem 9] for a proof.

**Lemma 2.7** (Post-processing property of Rényi divergence). *For any Rényi parameter $\alpha \geqslant 1$, any (possibly random) function $h$, and any probability distributions $\mu, \nu$,*

$$\mathcal{D}_\alpha \left( h_\# \mu \parallel h_\# \nu \right) \leqslant \mathcal{D}_\alpha \left( \mu \parallel \nu \right).$$

The third property is the appropriate analog of the chain rule for the KL divergence (the direct analog of the chain rule does not hold for Rényi divergences, $\alpha \neq 1$). This bound has appeared in various forms in previous work (e.g., [2, 15, 31]); we state the following two equivalent versions of this bound since at various points in our analysis it will be more convenient to reference one or the other. A proof of the former version is in [31, Proposition 3]. The proof of the latter is similar to [2, Theorem 2]; for the convenience of the reader, we provide a brief proof in Appendix A.1.

**Lemma 2.8** (Strong composition for RDP, v1). *Suppose algorithms $\mathcal{A}_1$ and $\mathcal{A}_2$ satisfy $(\alpha, \varepsilon_1)$-RDP and $(\alpha, \varepsilon_2)$-RDP, respectively. Let $\mathcal{A}$ denote the algorithm which, given a dataset $\mathcal{X}$ as input, outputs the composition $\mathcal{A}(\mathcal{X}) = (\mathcal{A}_1(\mathcal{X}), \mathcal{A}_2(\mathcal{X}, \mathcal{A}_1(\mathcal{X})))$. Then $\mathcal{A}$ satisfies $(\alpha, \varepsilon_1 + \varepsilon_2)$-RDP.*

**Lemma 2.9** (Strong composition for RDP, v2). *For any Rényi parameter $\alpha \geqslant 1$ and any two sequences of random variables $X_1, \ldots, X_k$ and $Y_1, \ldots, Y_k$,*

$$\mathcal{D}_\alpha \left( \mathbb{P}_{X_{1:k}} \parallel \mathbb{P}_{Y_{1:k}} \right) \leqslant \sum_{i=1}^k \sup_{x_{1:i-1}} \mathcal{D}_\alpha \left( \mathbb{P}_{X_i | X_{1:i-1} = x_{1:i-1}} \parallel \mathbb{P}_{Y_i | Y_{1:i-1} = x_{1:i-1}} \right).$$

## 2.3 Privacy Amplification by Sampling

A core technique in the DP literature is Privacy Amplification by Sampling [26] which quantifies the idea that a private algorithm, run on a small random sample of the input, becomes more private. There are several ways of formalizing this. For our purpose of analyzing NOISY-SGD, we must understand how distinguishable a noisy stochastic gradient update is when run on two adjacent datasets. This is precisely captured by the "Sampled Gaussian Mechanism", which is a composition of two operations: subsampling and additive Gaussian noise. Below we recall a convenient statement of this in terms of RDP from [32]. We start with a preliminary definition in one dimension.

**Definition 2.10** (Rényi Divergence of the Sampled Gaussian Mechanism). *For Rényi parameter $\alpha \geqslant 1$, mixing probability $q \in (0, 1)$, and noise parameter $\sigma > 0$, define*

$$S_\alpha(q, \sigma) := \mathcal{D}_\alpha\left(\mathcal{N}(0, \sigma^2) \,\big\|\, (1-q)\mathcal{N}(0, \sigma^2) + q\mathcal{N}(1, \sigma^2)\right).$$

Next, we provide a simple lemma that extends this notion to higher dimensions. In words, the proof simply argues that the worst-case distribution $\mu$ is a Dirac supported on a point of maximal norm, and then reduces the multivariate setting to the univariate setting via rotational invariance. The proof is similar to [32, Theorem 4]; for convenience, details are provided in Appendix A.2.

**Lemma 2.11** (Extrema for Rényi Divergence of the Sampled Gaussian Mechanism). *For any Rényi parameter $\alpha \geqslant 1$, mixing probability $q \in (0, 1)$, noise level $\sigma > 0$, dimension $d \in \mathbb{N}$, and radius $R > 0$,*

$$\sup_{\mu \in \mathcal{P}(R\mathbb{B}_d)} \mathcal{D}_\alpha\left(\mathcal{N}(0, \sigma^2 I_d) \,\big\|\, (1-q)\mathcal{N}(0, \sigma^2 I_d) + q\left(\mathcal{N}(0, \sigma^2 I_d) * \mu\right)\right) = S_\alpha(q, \sigma/R),$$

*where above $\mathcal{P}(R\mathbb{B}_d)$ denotes the set of Borel probability distributions that are supported on the ball of radius $R$ in $\mathbb{R}^d$.*

Finally, we recall the tight bound [32, Theorem 11] on these quantities. This bound restricts to Rényi parameter at most $\alpha^*(q, \sigma)$, which is defined to be the largest $\alpha$ satisfying $\alpha \leqslant M\sigma^2/2 - \log(\sigma^2)$ and $\alpha \leqslant (M^2\sigma^2/2 - \log(5\sigma^2))/(M + \log(q\alpha) + 1/(2\sigma^2))$, where $M = \log(1 + 1/(q(\alpha - 1)))$. While we use Lemma 2.12 to prove the asymptotics in our theoretical results, we emphasize that i) our bounds can be computed numerically for *any* $\alpha \geqslant 1$; and ii) this upper bound does not preclude $\alpha$ from the typical parameter regime of interest, see the discussion in §1.2.

**Lemma 2.12** (Bound on Rényi Divergence of the Sampled Gaussian Mechanism). *Consider Rényi parameter $\alpha > 1$, mixing probability $q \in (0, 1/5)$, and noise level $\sigma \geqslant 4$. If $\alpha \leqslant \alpha^*(q, \sigma)$, then*

$$S_\alpha(q, \sigma) \leqslant 2\alpha q^2/\sigma^2.$$

## 2.4 Privacy Amplification by Iteration

The Privacy Amplification by Iteration technique of [18] bounds the privacy loss of an iterative algorithm without "releasing" the entire sequence of iterates—unlike arguments based on Privacy Amplification by Sampling, c.f., the discussion in §1.3. This technique applies to processes which are generated by a Contractive Noisy Iteration (CNI). We begin by recalling this definition, albeit in a slightly more general form that allows for two differences. The first difference is allowing the contractions to be random; albeit simple, this generalization is critical for analyzing NOISY-SGD because a stochastic gradient update is random. The second difference is that we project each iterate; this generalization is solely for convenience as it simplifies the exposition.[8]

**Definition 2.13** (Contractive Noisy Iteration). *Consider a (random) initial state $X_0 \in \mathbb{R}^d$, a sequence of (random) contractions $\phi_t : \mathbb{R}^d \to \mathbb{R}^d$, a sequence of noise distributions $\xi_t$, and a convex set $\mathcal{K}$. The Projected Contractive Noisy Iteration $\mathrm{CNI}(X_0, \{\phi_t\}, \{\xi_t\}, \mathcal{K})$ is the law of the final iterate $X_T$ of the process*

$$X_{t+1} = \Pi_\mathcal{K}\left[\phi_{t+1}(X_t) + Z_{t+1}\right],$$

*where $Z_{t+1}$ is drawn independently from $\xi_{t+1}$.*

Privacy Amplification by Iteration is based on two key lemmas. We recall both below in the case of Gaussian noise since this suffices for our purposes. We begin with a preliminary definition.

**Definition 2.14** (Shifted Rényi Divergence; Definition 8 of [18]). *Let $\mu, \nu$ be probability distributions over $\mathbb{R}^d$. For parameters $z \geqslant 0$ and $\alpha \geqslant 1$, the shifted Rényi divergence is*

$$\mathcal{D}_\alpha^{(z)}(\mu \,\|\, \nu) = \inf_{\mu' : W_\infty(\mu, \mu') \leqslant z} D_\alpha(\mu' \,\|\, \nu).$$

---

[8]Since projections are contractive, these processes could be dubbed Contractive Noisy Contractive Processes (CNCI); however, we use Definition 2.13 as it more closely mirrors previous usages of CNI. Alternatively, since the composition of contractions is a contraction, the projection can be combined with $\phi_t$ in order to obtain a bona fide CNI; however, this requires defining auxiliary shifted processes which leads to a more complicated analysis (this was in the original arXiv version v1).

(Recall that the $\infty$-Wasserstein metric $\mathcal{W}_\infty(\mu, \mu')$ between two distributions $\mu$ and $\mu'$ is the smallest real number $w$ for which the following holds: there exists a joint distribution $\mathbb{P}_{X,X'}$ with first marginal $X \sim \mu$ and second marginal $X' \sim \mu'$, under which $\|X - X'\| \leqslant w$ almost surely.)

The shift-reduction lemma [18, Lemma 20] bounds the shifted Rényi divergence between two distributions that are convolved with Gaussian noise.

**Lemma 2.15** (Shift-reduction lemma). *Let $\mu, \nu$ be probability distributions on $\mathbb{R}^d$. For any $a \geqslant 0$,*

$$\mathcal{D}_\alpha^{(z)}\left(\mu * \mathcal{N}(0, \sigma^2 I_d) \,\|\, \nu * \mathcal{N}(0, \sigma^2 I_d)\right) \leqslant \mathcal{D}_\alpha^{(z+a)}\left(\mu \,\|\, \nu\right) + \frac{\alpha a^2}{2\sigma^2}.$$

The contraction-reduction lemma [18, Lemma 21] bounds the shifted Rényi divergence between the pushforwards of two distributions through similar contraction maps. Below we state a slight generalization of [18, Lemma 21] that allows for *random* contraction maps. The proof of this generalization is similar, except that here we exploit the quasi-convexity of the Rényi divergence to handle the additional randomness; details in Appendix A.3.

**Lemma 2.16** (Contraction-reduction lemma, for random contractions). *Suppose $\phi, \phi'$ are random functions from $\mathbb{R}^d$ to $\mathbb{R}^d$ such that (i) each is a contraction almost surely; and (ii) there exists a coupling of $(\phi, \phi')$ under which $\sup_z \|\phi(z) - \phi'(z)\| \leqslant s$ almost surely. Then for any probability distributions $\mu$ and $\mu'$ on $\mathbb{R}^d$,*

$$\mathcal{D}_\alpha^{(z+s)}\left(\phi_\# \mu \| \phi'_\# \mu'\right) \leqslant \mathcal{D}_\alpha^{(z)}\left(\mu \| \mu'\right).$$

The original[9] Privacy Amplification by Iteration argument combines these two lemmas to establish the following bound.

**Proposition 2.17** (Original PABI bound). *Let $X_T$ and $X'_T$ denote the outputs of $\mathrm{CNI}(X_0, \{\phi_t\}, \{\xi_t\}, \mathcal{K})$ and $\mathrm{CNI}(X_0, \{\phi'_t\}, \{\xi_t\}, \mathcal{K})$ where $\xi_t = \mathcal{N}(0, \sigma_t^2 I_d)$. Let $s_t := \sup_x \|\phi_t(x) - \phi'_t(x)\|$, and consider any sequence $a_1, \ldots, a_T$ such that $z_t := \sum_{i=1}^t (s_i - a_i)$ is non-negative for all $t$ and satisfies $z_T = 0$. Then*

$$\mathcal{D}_\alpha\left(\mathbb{P}_{X_T} \,\|\, \mathbb{P}_{X'_T}\right) \leqslant \frac{\alpha}{2} \sum_{t=1}^T \frac{a_t^2}{\sigma_t^2}.$$

# 3 Upper bound on privacy

In this section, we prove the upper bound in Theorem 1.3. The formal statement of this result is as follows; see §1.2 for a discussion of the mild assumptions on $\sigma$ and $\alpha$.

**Theorem 3.1** (Privacy upper bound for NOISY-SGD). *Let $\mathcal{K} \subset \mathbb{R}^d$ be a convex set of diameter $D$, and consider optimizing convex losses over $\mathcal{K}$ that are $L$-Lipschitz and $M$-smooth. For any number of iterations $T$, dataset size $n \in \mathbb{N}$, batch size $b \leqslant n$, stepsize $\eta \leqslant 2/M$, noise parameter $\sigma > 8\sqrt{2}L/b$, and initialization $\omega_0 \in \mathcal{K}$, NOISY-SGD satisfies $(\alpha, \varepsilon)$-RDP for $1 < \alpha \leqslant \alpha^*\left(\frac{b}{n}, \frac{b\sigma}{2\sqrt{2}L}\right)$ and*

$$\varepsilon \lesssim \frac{\alpha L^2}{n^2 \sigma^2} \min\left\{T, \bar{T}\right\},$$

*where $\bar{T} := \lceil \frac{Dn}{L\eta} \rceil$.*

Below, in §3.1 we first isolate a simple ingredient in our analysis as it may be of independent interest. Then in §3.2 we prove Theorem 3.1.

## 3.1 Privacy Amplification by Iteration bounds that are not vacuous as $T \to \infty$

Recall from the preliminaries section §2.4 that Privacy Amplification by Iteration arguments, while tight for a small number of iterations $T$, provide vacuous bounds as $T \to \infty$ (c.f., Proposition 2.17).

---

[9]Strictly speaking, Proposition 2.17 is a generalization of [18, Theorem 22] since it allows for randomized contractions and projections in the CNI (c.f., Definition 2.13). However, the proof is identical, modulo replacing the original Contraction-Reduction Lemma with its randomized generalization (Lemma 2.16) and analyzing the projection step again using the Contraction-Reduction Lemma.

The following proposition overcomes this by establishing privacy bounds which are *independent* of the number of iterations $T$. This result only requires additionally assuming that $\|X_\tau - X'_\tau\|$ is bounded at some intermediate time $\tau$. This is a mild assumption that is satisfied automatically if, e.g., both CNI processes are in a constraint set of bounded diameter.

**Proposition 3.2** (New PABI bound that is not vacuous as $T \to \infty$). *Let $X_T$, $X'_T$, and $s_t$ be as in Proposition 2.17. Consider any $\tau \in \{0, \ldots, T-1\}$ and sequence $a_{\tau+1}, \ldots, a_T$ such that $z_t := D + \sum_{i=\tau+1}^{t}(s_i - a_i)$ is non-negative for all $t$ and satisfies $z_T = 0$. If $\mathcal{K}$ has diameter $D$, then:*

$$\mathcal{D}_\alpha \left( \mathbb{P}_{X_T} \parallel \mathbb{P}_{X'_T} \right) \leqslant \frac{\alpha}{2} \sum_{t=\tau+1}^{T} \frac{a_t^2}{\sigma_t^2}.$$

In words, the main idea behind Proposition 3.2 is simply to change the original Privacy Amplification by Iteration argument—which bounds the shifted divergence at iteration $T$, by the shifted divergence at iteration $T-1$, and so on all the way to the shifted divergence at iteration $0$—by instead stopping the induction earlier. Specifically, only unroll to iteration $\tau$, and then use boundedness of the iterates to control the shifted divergence at that intermediate time $\tau$.

We remark that this new version of PABI uses the shift in the shifted Rényi divergence for a different purpose than previous work: rather than just using the shift to bound the bias incurred from updating on two different losses, here we also use the shift to exploit the boundedness of the constraint set.

*Proof of Proposition 3.2.* Bound the divergence at iteration $T$ by the shifted divergence at iteration $T-1$ as follows:

$$\begin{aligned}
\mathcal{D}_\alpha \left( \mathbb{P}_{X_T} \parallel \mathbb{P}_{X'_T} \right) &= \mathcal{D}_\alpha^{(z_T)} \left( \mathbb{P}_{X_T} \parallel \mathbb{P}_{X'_T} \right) \\
&= \mathcal{D}_\alpha^{(z_{T-1}+s_T-a_T)} \left( \mathbb{P}_{\Pi_\mathcal{K}[\phi_T(X_{T-1})+Z_T]} \parallel \mathbb{P}_{\Pi_\mathcal{K}[\phi'_{T-1}(X'_{T-1})+Z'_T]} \right) \\
&\leqslant \mathcal{D}_\alpha^{(z_{T-1}+s_T-a_T)} \left( \mathbb{P}_{\phi_T(X_{T-1})+Z_T} \parallel \mathbb{P}_{\phi'_{T-1}(X'_{T-1})+Z'_T} \right) \\
&\leqslant \mathcal{D}_\alpha^{(z_{T-1}+s_T)} \left( \mathbb{P}_{\phi_T(X_{T-1})} \parallel \mathbb{P}_{\phi'_{T-1}(X'_{T-1})} \right) + \frac{\alpha a_T^2}{2\sigma_T^2} \\
&\leqslant \mathcal{D}_\alpha^{(z_{T-1})} \left( \mathbb{P}_{X_{T-1}} \parallel \mathbb{P}'_{X_{T-1}} \right) + \frac{\alpha a_T^2}{2\sigma_T^2}.
\end{aligned}$$

Above, the first step is because $z_T = 0$; the second step is by the iterative construction of $X_T, X'_T, z_T$; the third and final steps are by the the contraction-reduction lemma (Lemma 2.16), and the penulimate step is by the shift-reduction lemma (Lemma 2.15.

By repeating the above argument, from $T$ to $T-1$ all the way to $\tau$, we obtain:

$$\mathcal{D}_\alpha \left( \mathbb{P}_{X_T} \parallel \mathbb{P}_{X'_T} \right) \leqslant \mathcal{D}_\alpha^{(z_\tau)} \left( \mathbb{P}_{X_\tau} \parallel \mathbb{P}'_{X_\tau} \right) + \frac{\alpha}{2} \sum_{t=\tau+1}^{T} \frac{a_t^2}{\sigma_t^2}.$$

Now observe that the shifted Rényi divergence on the right hand side vanishes because $z_\tau = D$. $\qquad\square$

## 3.2 Proof of Theorem 3.1

**Step 1: Coupling the iterates**

Suppose $\mathcal{X} = \{x_1, \ldots, x_n\}$ and $\mathcal{X}' = \{x'_1, \ldots, x'_n\}$ are adjacent datasets; that is, they agree $x_i = x'_i$ on all indices $i \in [n] \setminus \{i^*\}$ except for at most one index $i^* \in [n]$. Denote the corresponding loss functions by $f_i$ and $f'_i$, where $f_i = f'_i$ except possibly $f_{i^*} \neq f'_{i^*}$. Consider running NOISY-SGD on either dataset $\mathcal{X}$ or $\mathcal{X}'$ for $T$ iterations—call the resulting iterates $\{W_t\}_{t=0}^T$ and $\{W'_t\}_{t=0}^T$, respectively—where we start from the same point $w_0 \in \mathcal{K}$ and couple the sequence of random batches $\{B_t\}_{t=0}^{T-1}$ and the random noise injected in each iteration. That is,

$$W_{t+1} = \Pi_\mathcal{K} \left[ W_t - \frac{\eta}{b} \sum_{i \in B_t} \nabla f_i(W_t) + Y_t + Z_t \right]$$

$$W'_{t+1} = \Pi_\mathcal{K} \left[ W'_t - \frac{\eta}{b} \sum_{i \in B_t} \nabla f_i(W'_t) + Y_t + Z'_t \right]$$

for all $t \in \{0, \ldots, T-1\}$, where we have split the Gaussian noise into terms $Y_t \sim \mathcal{N}(0, \eta^2 \sigma_1^2 I_d)$, $Z_t \sim \mathcal{N}(0, \eta^2 \sigma_2^2 I_d)$, $Z'_t \sim \mathcal{N}(0, \eta^2 \sigma_2^2 I_d) + \frac{\eta}{b} [\nabla f'_{i^*}(W'_t) - \nabla f_{i^*}(W'_t)] \cdot \mathbb{1}_{i^* \in B_t}$, for any numbers $\sigma_1, \sigma_2 > 0$ satisfying $\sigma_1^2 + \sigma_2^2 = \sigma^2$. (We set $\sigma_1 = \sigma_2 = \sigma/\sqrt{2}$ later for asymptotics.) In words, this noise-splitting enables us to use the noise for both the Privacy Amplification by Sampling and Privacy Amplification by Iteration arguments below.

Importantly, notice that in the definition of $W'_{t+1}$, the gradient is taken w.r.t. loss functions corresponding to data set $\mathcal{X}$ rather than $\mathcal{X}'$; this is then corrected via the bias in the noise term $Z'_t$. Notice also that this bias term in $Z'_t$ is only realized (i.e., $Z'_t$ is possibly non-centered) with probability $1 - b/n$ because the probability that $i^*$ is in a random size-$b$ subset of $[n]$ is

$$\mathbb{P}[i^* \in B_t] = \frac{b}{n}. \tag{3.1}$$

**Step 2: Interpretation as conditional CNI sequences**

Observe that conditional on the event that $Z_t = Z'_t$ are equal (call their value $z_t$), then

$$W_{t+1} = \Pi_{\mathcal{K}} [\phi_t(W_t) + Y_t]$$
$$W'_{t+1} = \Pi_{\mathcal{K}} [\phi_t(W'_t) + Y_t]$$

where

$$\phi_t(\omega) := \omega - \frac{\eta}{b} \sum_{i \in B_t} \nabla f_i(\omega) + z_t. \tag{3.2}$$

Since the following lemma establishes that $\phi_t$ is contractive, we conclude that conditional on the event that $Z_t = Z'_t$ for all $t \geqslant \tau$, then $\{W_t\}_{t \geqslant \tau}$ and $\{W'_t\}_{t \geqslant \tau}$ are projected CNI (c.f., Definition 2.13) with respect to the same update functions. Here, $\tau \in \{0, \ldots, T-1\}$ is a parameter that will be chosen shortly. Intuitively, $\tau$ is the horizon for which we bound all previous privacy leakage only through the fact that $W_\tau, W'_\tau$ are within distance $D$ from each other, see the proof overview in §1.3.

**Observation 3.3.** *The function $\phi_t$ defined in (3.2) is contractive.*

*Proof.* For any $\omega, \omega'$,

$$\|\phi_t(\omega) - \phi_t(\omega')\| = \left\| \left(\omega - \frac{\eta}{b} \sum_{i \in B_t} \nabla f_i(\omega)\right) - \left(\omega' - \frac{\eta}{b} \sum_{i \in B_t} \nabla f_i(\omega')\right) \right\|$$

$$\leqslant \frac{1}{b} \sum_{i \in B_t} \|(\omega - \eta \nabla f_i(\omega)) - (\omega' - \eta \nabla f_i(\omega'))\|$$

$$\leqslant \frac{1}{b} \sum_{i \in B_t} \|\omega - \omega'\|$$

$$= \|\omega - \omega'\|.$$

by plugging in the definition of $\phi_t$, using the triangle inequality, and then using the fact that the stochastic gradient update $\omega \mapsto \omega - \eta \nabla f_i(\omega)$ is a contraction (Lemma 2.1). $\qquad \square$

**Step 3: Bounding the privacy loss**

Recall that we seek to upper bound $\mathcal{D}_\alpha(\mathbb{P}_{W_T} \| \mathbb{P}_{W'_T})$. We argue that:

$$\mathcal{D}_\alpha \left( \mathbb{P}_{W_T} \| \mathbb{P}_{W'_T} \right) \leqslant \mathcal{D}_\alpha \left( \mathbb{P}_{W_T, Z_{\tau:T-1}} \| \mathbb{P}_{W'_T, Z'_{\tau:T-1}} \right)$$

$$\leqslant \underbrace{\mathcal{D}_\alpha \left( \mathbb{P}_{Z_{\tau:T-1}} \| \mathbb{P}_{Z'_{\tau:T-1}} \right)}_{\textcircled{1}} + \underbrace{\sup_z \mathcal{D}_\alpha \left( \mathbb{P}_{W_T | Z_{\tau:T-1} = z} \| \mathbb{P}_{W'_T | Z'_{\tau:T-1} = z} \right)}_{\textcircled{2}} \tag{3.3}$$

Above, the first step is by the post-processing inequality for the Rényi divergence (Lemma 2.7), and the second step is by the strong composition rule for the Rényi divergence (Lemma 2.9).

Step 3a: Bounding $\boxed{1}$, using Privacy Amplification by Sampling. We argue that

$$
\boxed{1} = \mathcal{D}_\alpha \left( \mathbb{P}_{Z_{\tau:T-1}} \,\|\, \mathbb{P}_{Z'_{\tau:T-1}} \right)
$$

$$
\leqslant \sum_{t=\tau}^{T-1} \sup_{z_{\tau:t-1}} \mathcal{D}_\alpha \left( \mathbb{P}_{Z_t | Z_{\tau:t-1} = z_{\tau:t-1}} \,\|\, \mathbb{P}_{Z'_t | Z'_{\tau:t-1} = z_{\tau:t-1}} \right)
$$

$$
= \sum_{t=\tau}^{T-1} \mathcal{D}_\alpha \left( \mathcal{N}(0, \eta^2 \sigma_2^2 I_d) \,\|\, (1 - \tfrac{b}{n}) \mathcal{N}(0, \eta^2 \sigma_2^2 I_d) + \tfrac{b}{n} \mathcal{N}(m_t, \eta^2 \sigma_2^2 I_d) \right)
$$

$$
\leqslant (T - \tau) S_\alpha \left( \frac{b}{n}, \frac{b\sigma_2}{2L} \right). \tag{3.4}
$$

Above, the first step is the definition of $\boxed{1}$. The second step is by the strong composition rule for the Rényi divergence (Lemma 2.9). The third step is because for any $z_{\tau:t-1}$, the law of $Z_t$ conditional on $Z_{\tau:t-1} = z_{\tau:t-1}$ is the Gaussian distribution $\mathcal{N}(0, \eta^2 \sigma_2^2 I_d)$; and by (3.1), the law of $Z'_t$ conditional on $Z_{\tau:t-1} = z_{\tau:t-1}$ is the mixture distribution that is $\mathcal{N}(0, \eta^2 \sigma_2^2 I_d)$ with probability $1 - b/n$, and otherwise is $\mathcal{N}(m_t, \eta^2 \sigma_2^2 I_d)$ where $m_t := \frac{\eta}{b} [\nabla f_{i^*}(W'_t) - \nabla f'_{i^*}(W'_t)]$. The final step is by the bound in Lemma 2.11 on the Rényi divergence of the Sampled Gaussian Mechanism, combined with the observation that $\|m_t\| \leqslant 2\eta L/b$, which is immediate from the triangle inequality and the $L$-Lipschitz smoothness of the loss functions.

Step 3b: Bounding $\boxed{2}$, using Privacy Amplification by Iteration. As argued in step 2, $\{W_t\}_{t \geqslant \tau}$ and $\{W'_t\}_{t \geqslant \tau}$ are projected CNI with respect to the same update functions conditional on the event that $Z_t = Z'_t$ for all $t \geqslant \tau$. Note also that $\|W_\tau - W_{\tau'}\| \leqslant D$ since the iterates lie in the constraint set $\mathcal{K}$ which has diameter $D$. Therefore we may apply the new Privacy Amplification by Iteration bound (Proposition 3.2) with $s_t \equiv 0$ and $a_t \equiv D/(T - \tau)$ to obtain:

$$
\boxed{2} = \sup_z \mathcal{D}_\alpha \left( \mathbb{P}_{W_T | Z_{\tau:T-1} = z} \,\|\, \mathbb{P}_{W'_T | Z'_{\tau:T-1} = z} \right) \leqslant \frac{\alpha D^2}{2\eta^2 \sigma_1^2 (T - \tau)}. \tag{3.5}
$$

**Step 4: Putting the bounds together**

By plugging into (3.3) the bound (3.4) on $\boxed{1}$ and the bound (3.5) on $\boxed{2}$, we conclude that the algorithm is $(\alpha, \varepsilon)$-RDP for

$$
\varepsilon \leqslant \min_{\tau \in \{0, \dots, T-1\}} \left\{ (T - \tau) S_\alpha \left( \frac{b}{n}, \frac{b\sigma_2}{2L} \right) + \frac{\alpha D^2}{2\eta^2 \sigma_1^2 (T - \tau)} \right\} \tag{3.6}
$$

By Lemma 2.12, $S_\alpha(\frac{b}{n}, \frac{b\sigma_2}{2L}) \leqslant 8\alpha(\frac{L}{n\sigma_2})^2$ for $\alpha \leqslant \alpha^*(\frac{b}{n}, \frac{b\sigma_2}{2L})$ and $\sigma_2 \geqslant 8L/b$.[10]

By setting $\sigma_1 = \sigma_2 = \sigma/\sqrt{2}$, we have that up to a constant factor,

$$
\varepsilon \lesssim \frac{\alpha L^2}{\sigma^2} \min_{\tau \in \{0, \dots, T-1\}} \left\{ \frac{(T - \tau)}{n^2} + \frac{D^2}{\eta^2 L^2 (T - \tau)} \right\}.
$$

Bound this minimization by

$$
\min_{\tau \in \{0, \dots, T-1\}} \left\{ \frac{(T - \tau)}{n^2} + \frac{D^2}{\eta^2 L^2 (T - \tau)} \right\} = \min_{R \in \{1, \dots, T\}} \left\{ \frac{R}{n^2} + \frac{D^2}{\eta^2 L^2 R} \right\} \lesssim \frac{D}{\eta L n},
$$

where above the first step is by setting $R = T - \tau$, and the second step is by setting $R = \bar{T} = \lceil \frac{Dn}{L\eta} \rceil$ (this can be done if $T \gtrsim \bar{T}$). Therefore, by combining the above two displays, we obtain

$$
\varepsilon \lesssim \frac{\alpha L^2}{n^2 \sigma^2} \min \left\{ T, \bar{T} \right\}.
$$

Here the first term in the minimization comes from the simple bound (1.1) which scales linearly in $T$. This completes the proof of Theorem 3.1.

---

[10] While Lemma 2.12 requires $b < n/5$, the case $b \geqslant n/5$ has an alternate proof that does not require Lemma 2.12 (or any of its assumptions). Specifically, replace (3.4) with the upper bound $\mathcal{D}_\alpha(\mathcal{N}(0, \eta^2 \sigma_2^2 I_d) \,\|\, \mathcal{N}(m_t, \eta^2 \sigma_2^2 I_d)) = \alpha \|m_t\|^2 / (2\eta^2 \sigma_2^2) = 2\alpha L^2 / b^2 \sigma_2^2$ by using the well-known formula for the Rényi divergence between Gaussians. This is tight up to a constant factor, and the rest of the proof proceeds identically.

# 4 Lower bound on privacy

In this section, we prove the lower bound in Theorem 1.3. This is stated formally below and holds even for linear loss functions in one dimension, in fact even when all but one of the loss functions are zero. See §1.2 for a discussion of the mild assumptions on the diameter.

**Theorem 4.1** (Privacy lower bound for NOISY-SGD). *There exist universal constants[11] $c_D$, $c_\sigma$, $c_\alpha$, $\bar{\alpha}$ and a family of $L$-Lipschitz linear loss functions over the interval $\mathcal{K} = [-D/2, D/2] \subset \mathbb{R}$ such that the following holds. Consider running NOISY-SGD from arbitrary initialization $\omega_0$ with any parameters satisfying $D \geqslant c_D \eta L$ and $\sigma^2 \leqslant c_\sigma D^2/(\eta^2 \bar{T})$. Then NOISY-SGD is not $(\bar{\alpha}, \varepsilon)$-RDP for*

$$\varepsilon \leqslant c_\alpha \frac{\bar{\alpha} L^2}{n^2 \sigma^2} \min\{T, \bar{T}\}, \tag{4.1}$$

*where $\bar{T} := 0.75 \frac{Dn}{L\eta}$.*

**Proof sketch of Theorem 4.1.** (Full details in Appendix A.4.)

Construction. Consider datasets $\mathcal{X} = \{x_1, \ldots, x_{n-1}, x_n\}$ and $\mathcal{X}' = \{x_1, \ldots, x_{n-1}, x'_n\}$ which differ only on $x'_n$, and corresponding functions which are all zero $f_1(\cdot) = \cdots = f_n(\cdot) = 0$, except for $f'_n(\omega) = L(D - \omega)$. Clearly these functions are linear and $L$-Lipschitz. The intuition behind this construction is that running NOISY-SGD on $\mathcal{X}$ or $\mathcal{X}'$ generates a random walk that is clamped to stay within the interval $\mathcal{K}$—but with the key difference that running NOISY-SGD on dataset $\mathcal{X}$ generates a *symmetric* random walk $\{\omega_t\}$, whereas running NOISY-SGD on dataset $\mathcal{X}'$ generates a *biased* random walk $\{\omega'_t\}$ that biases right with probability $b/n$ each step. That is,

$$\omega_{t+1} = \Pi_{\mathcal{K}}\left[\omega_t + Z_t\right] \qquad \text{and} \qquad \omega'_{t+1} = \Pi_{\mathcal{K}}\left[\omega'_t + Y_t + Z_t\right]$$

where the processes are initialized at $\omega_0 = \omega'_0 = 0$, each random increment $Z_t \sim \mathcal{N}(0, \eta^2 \sigma^2)$ is an independent Gaussian, and each bias $Y_t$ is $\eta L/b$ with probability $b/n$ and otherwise is $0$.

Key obstacle. The high-level intuition behind this construction is simple to state: the bias of the random walk $\{\omega'_t\}$ makes it distinguishable (to the minimax-optimal extent, as we show) from the symmetric random walk $\{\omega_t\}$. However, making this intuition precise is challenging because the distributions of the iterates $\omega_t, \omega'_t$ are intractable to reason about explicitly—due to the highly non-linear interactions between the projections and the random increments. Thus we must establish the distinguishability of $\omega_T, \omega'_T$ without reasoning explicitly about their distributions.

Key technical ideas. A first, simple observation that it suffices to prove Theorem 4.1 in the constant RDP regime of $T \geqslant \bar{T}$, since the linear RDP regime of $T \leqslant \bar{T}$ then follows by the strong composition rule for RDP (Lemma 2.8), as described in Appendix A.4. Thus it suffices to show that the final iterates $\omega_T, \omega'_T$ are distinguishable after $T \geqslant \bar{T}$ iterations.

A natural attempt to distinguish $\omega_T, \omega'_T$ for large $T$ is to test positivity. This is intuitively plausible because $\mathbb{P}[\omega_T \geqslant 0] = 1/2$ by symmetry, whereas we might expect $\mathbb{P}[\omega'_T \geqslant 0] \gg 1/2$ since the bias of $\omega'_T$ pushes it to the top half of the interval $\mathcal{K} = [-D/2, D/2]$. Such a discrepancy would establish an $(\varepsilon, \delta)$-DP bound that, by the standard RDP-to-DP-conversion in Lemma 2.5, would imply the desired $(\alpha, \varepsilon)$-RDP bound in Theorem 4.1. However, the issue is how to prove the latter statement $\mathbb{P}[\omega'_T \geqslant 0] \gg 1/2$ without an explicit handle on the distribution of $\omega'_T$.

To this end, the key technical insight is to define an auxiliary process $\omega''_t$ which intializes $\bar{T}$ iterations before the final iteration $T$ at the lowest point in $\mathcal{K}$, namely $\omega''_{T-\bar{T}} = -D/2$, and then updates in an analogously biased way as the $\omega'_t$ process except without projections at the bottom of $\mathcal{K}$. That is,

$$\omega''_{t+1} := \min\left(\omega''_t + Y_t + Z_t, D/2\right).$$

The point is that on one hand, $\omega'_t$ stochastically dominates $\omega''_t$, so that it suffices to show $\mathbb{P}[\omega''_T \geqslant 0] \gg 1/2$. And on the other hand, $\omega''_t$ is easy to analyze because, as we show, with overwhelming probability no projections occur. This lack of projections means that, modulo a probability $\delta$ event which is irrelevant for the purpose of $(\varepsilon, \delta)$-DP bounds, $\omega''_T$ is the sum of (biased) independent Gaussian increments—hence it has a simple explicitly computable distribution: it is Gaussian.

---

[11]We prove this for $c_\sigma = 10^{-3}$, $c_D = 10^3$, $c_\alpha = 10^{-7}$, $\bar{\alpha} = 10^2$; no attempt has been made to optimize these constants.

It remains to establish that (i) with high probability no projections occur in the $\omega_t''$ process, so that the aforementioned Gaussian approximates the law of $\omega_T''$, and (ii) this Gaussian is positive with probability $\gg 1/2$, so that we may conclude the desired DP lower bound. Briefly, the former amounts to bounding the hitting time of a Gaussian random walk, which is a routine application of martingale concentration. And the latter amounts to computing the parameters of the Gaussian. A back-of-the-envelope calculation shows that the total bias in the $\omega_t''$ process is $\sum_{t=T-\bar{T}}^{T-1} Y_t \approx \bar{T}\eta L/n = 3D/4$, and conditional on such an event, the Gaussian approximating $\omega_T''$ has mean roughly $-D/2 + 3D/4 = D/4$ and variance $\bar{T}\eta^2\sigma^2 \leqslant D^2/1000$, and therefore is positive with probability $\gg 1/2$, as desired. Full proof details are provided in Appendix A.4.

## 5 Extension to the strongly convex setting

Here we describe how our techniques readily extend to strongly convex losses. In particular, we show that after a much lower threshold of $\tilde{O}(\kappa)$ iterations, there is no further privacy loss. Throughout, the notation $\tilde{O}$ suppresses logarithmic factors in the relevant parameters.

**Theorem 5.1** (Privacy upper bound for NOISY-SGD, in strongly convex setting)**.** *Let $\mathcal{K} \subset \mathbb{R}^d$ be a convex set of diameter $D$, and consider optimizing losses over $\mathcal{K}$ that are $L$-Lipschitz, $m$-strongly convex, and $M$-smooth. Denote the condition number by $\kappa := M/m \geqslant 1$. For any number of iterations $T$, dataset size $n \in \mathbb{N}$, batch size $b \leqslant n$, noise parameter $\sigma > 8\sqrt{2}L/b$, and initialization $\omega_0 \in \mathcal{K}$, NOISY-SGD with stepsize $\eta = 2/(M+m)$ satisfies $(\alpha, \varepsilon)$-RDP for $1 < \alpha < \alpha^*(\frac{b}{n}, \frac{b\sigma}{2\sqrt{2}L})$ and*

$$\varepsilon \lesssim \frac{\alpha L^2}{n^2\sigma^2} \cdot \min\{T, \bar{T}\}, \tag{5.1}$$

*where $\bar{T} = \tilde{O}(\kappa)$.*

We make two remarks about this result.

**Remark 5.2** (Bounded diameter is unnecessary for convergent privacy in the strongly convex setting)**.** *Unlike the convex setting, in this strongly convex setting, the bounded diameter assumption can be removed. Specifically, for the purposes of DP (rather than RDP), the logarithmic dependence of $\bar{T}$ on $D$ (hidden in the $\tilde{O}$ above) can be replaced by logarithmic dependence on $T$, $\eta$, $L$, and $\sigma$ since the SGD trajectories move from the initialization point by $O(T\eta(L+\sigma))$ with high probability, and so this can act as the "effective diameter".*

**Remark 5.3** (General stepsizes)**.** *For simplicity, Theorem 5.1 is stated for $\eta = 2/(M+m)$. The same argument and result apply for other stepsizes $\eta < 2/M$, with the $\kappa$ factor in the final bound replaced by $1/\log(1/c)$, where $c = \max_{\lambda \in \{m,M\}} |1 - \eta\lambda|$. The point is that this is more generally the contraction coefficient for gradient descent with non-optimized stepsize (c.f., Lemma 5.5). For example, if $\eta < 1/M$, then $c = 1 - \eta m \leqslant \exp(-\eta m)$, whereby the $\kappa$ factor is replaced by $1/(\eta m)$.*

The proof of Theorem 5.1 (the setting of strongly convex losses) is similar to the proof of Theorem 3.1 (the setting of convex losses), except for two key differences:

(i) Strong convexity of the losses ensures that the update function $\phi_t$ in the Contractive Noisy Iterations is *strongly* contractive, i.e., $c$-contractive for $c < 1$. (Formalized in Observation 5.4.)

(ii) This strong contractivity of the update function ensures exponentially better bounds in the Privacy Amplification by Iteration argument. (Formalized in Proposition 5.6.)

We first formalize the change (i).

**Observation 5.4** (Analog of Observation 3.3 for strongly convex losses)**.** *For all $t$, the function $\phi_t$ defined in (3.2) is almost surely $c$-contractive, for $c = (\kappa - 1)/(\kappa + 1)$.*

*Proof.* Identical to the proof of Observation 3.3, except use the fact that a gradient step is not simply contractive (Lemma 2.1), but in fact strongly contractive when the function is strongly convex (this fact is recalled in Lemma 5.5 below; see, e.g., [8, Theorem 3.12] for a proof). $\square$

**Lemma 5.5** (Analog of Lemma 2.1 for strongly convex losses)**.** *Suppose $f : \mathbb{R}^d \to \mathbb{R}$ is an $m$-strongly convex, $M$-smooth function. For stepsize $\eta = 2/(M+m)$, the mapping $\omega \mapsto \omega - \eta\nabla f(\omega)$ is $c$-contractive, for $c = (\kappa - 1)/(\kappa + 1)$.*

Next we formalize the change (ii).

**Proposition 5.6** (Analog of Proposition 3.2 for strongly convex losses). *Consider the setup in Proposition 3.2, and additonally assume that $\phi_t, \phi'_t$ are almost surely $c$-contractive. Consider any $\tau \in \{0, \dots, T-1\}$ and any reals $a_{\tau+1}, \dots, a_T$ such that $z_t := c^{t-\tau} D + \sum_{i=\tau+1}^t c^{t-i}(s_i - a_i)$ is non-negative for all $t$ and satisfies $z_T = 0$. Then:*

$$\mathcal{D}_\alpha \left( \mathbb{P}_{X_T} \| \mathbb{P}_{X'_T} \right) \leqslant \frac{\alpha}{2} \sum_{t=\tau+1}^T \frac{a_t^2}{\sigma_t^2}.$$

*Proof.* Identical to the proof of Proposition 3.2, except use the contraction-reduction lemma for strong contractions (Lemma 5.7 below rather than Lemma 2.16), and the inductive relation $z_{t+1} = cz_t + s_{t+1} - a_{t+1}$. $\qquad \square$

**Lemma 5.7** (Analog of Lemma 2.16 for strongly convex losses). *Consider the setup of Lemma 5.7, except additionally assuming that $\phi, \phi'$ are each $c$-contractive almost surely. Then*

$$\mathcal{D}_\alpha^{(cz+s)} \left( \phi_\# \mu \| \phi'_\# \mu' \right) \leqslant \mathcal{D}_\alpha^{(z)} \left( \mu \| \mu' \right).$$

*Proof.* Identical to the proof of Lemma 2.16, except use the fact that $\phi$ is almost surely a $c$-contraction to bound the second term in (A.1), namely by $W_\infty(\phi_\# \nu, \phi_\# \mu) \leqslant c W_\infty(\nu, \mu) \leqslant cz$. $\qquad \square$

We now combine the changes (i) and (ii) to prove Theorem 5.1.

*Proof of Theorem 5.1.* We record the differences to Steps 1-4 of the proof in the convex setting (Theorem 3.1). Step 1 is identical for coupling the iterates. Step 2 is identical for constructing the conditional CNI, except that in this strongly convex setting, the update functions $\phi_t$ are not simply contractive (Observation 3.3), but in fact $c$-contractive (Observation 5.4) for $c = (\kappa - 1)/(\kappa + 1)$. We use this to establish an improved bound on the term ②️ in Step 3. Specifically, invoke Proposition 5.6 with $s_t = 0$ for all $t \in \{\tau + 1, \dots, T\}$, $a_t = 0$ for all $t \in \{\tau + 1, \dots, T-1\}$ and $a_T = c^{T-\tau} D$ to obtain

$$\text{②️} = \sup_z \mathcal{D}_\alpha \left( \mathbb{P}_{W_T | Z_{\tau:T-1} = z} \| \mathbb{P}_{W'_T | Z'_{\tau:T-1} = z} \right) \leqslant c^{2(T-\tau)} \frac{\alpha D^2}{2\eta^2 \sigma_1^2}. \tag{5.2}$$

This allows us to conclude the following analog of (3.6) for the present strongly convex setting:

$$\varepsilon \leqslant \min_{\tau \in \{0, \dots, T-1\}} (T - \tau) Q + c^{2(T-\tau)} \frac{\alpha D^2}{2\eta^2 \sigma_1^2}, \tag{5.3}$$

where $Q := S_\alpha(\frac{b}{n}, \frac{b\sigma_2}{2L})$. Simplifying this in Step 4 requires the following modifications since the asymptotics are different in the present strongly convex setting. Specifically, bound the above by

$$\varepsilon \lesssim Q\kappa \log \left( \frac{\alpha D^2}{\eta^2 \sigma_1^2 Q\kappa} \right).$$

by bounding $c = (\kappa - 1)/(\kappa + 1) \leqslant 1 - 1/\kappa \leqslant e^{-1/\kappa}$ and by setting $\tau = T - \Theta(\kappa \log((\alpha D^2)/(\eta^2 \sigma_1^2 Q\kappa))) = T - \tilde{\Theta}(\kappa)$, which is valid if $\tau \geqslant 0$. By setting $\sigma_1 = \sigma_2 = \sigma/\sqrt{2}$, and using the bound on $Q$ in the proof of Theorem 3.1, we conclude the desired bound

$$\varepsilon \lesssim \frac{\alpha L^2}{n^2 \sigma^2} \min \left\{ T, \kappa \log \left( \frac{\alpha D^2}{\eta^2 \sigma_1^2 Q\kappa} \right) \right\} = \frac{\alpha L^2}{n^2 \sigma^2} \cdot \min \left\{ T, \tilde{O}(\kappa) \right\}.$$

$\qquad \square$

## 6   Discussion

The results of this paper suggest several natural directions for future work:

Clipped gradients? In practical settings, NOISY-SGD implementations sometimes "clip" gradients to force their norms to be small, see e.g., [2]. In the case of generalized linear models, the clipped gradients can be viewed as gradients of an auxiliary convex loss [39], in which case our results

can be applied directly. However, in general, clipped gradients do not correspond to gradients of a convex loss, in which case our results (as well as all other works in the literature that aim at proving convergent privacy bounds) do not apply. Can this be remedied?

Average iterate? Can similar privacy guarantees be established for the average iterate rather than the last iterate? There are fundamental difficulties with trying to proving this: indeed, the average iterate is provably not as private for NOISY-CSGD [5].

Adaptive stepsizes? Can similar privacy guarantees be established for optimization algorithms with adaptive stepsizes? The main technical obstacle is how to control the privacy loss from how past iterates affect the adaptivity in later iterates. This appears to preclude using our analysis techniques, at least in their current form.

Beyond convexity? Convergent privacy bounds break down without convexity. This precludes applicability to deep neural networks. Is there any hope of establishing similar results under some sort of mild non-convexity? Due to simple non-convex counterexamples where the privacy of NOISY-SGD diverges, any such extension would have to make additional structural assumptions on the non-convexity (and also possibly change the NOISY-SGD algorithm), although it is unclear how this would even look. Moreover, this appears to require significant new machinery as our techniques are the only known way to solve the convex problem, and they break down in the non-convex setting (see also the discussion in §1.2).

General techniques? Can the analysis techniques developed in this paper be used in other settings? Our techniques readily generalize to any iterative algorithm which interleaves contractive steps and noise convolutions. Such algorithms are common in differentially private optimization, and it would be interesting to apply them to variants of NOISY-SGD.

## Acknowledgements.

We are grateful to Hristo Paskov for many insightful conversations.

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

# A  Deferred proofs

## A.1  Proof of Lemma 2.9

For $\alpha > 1$,

$$
\begin{aligned}
\exp\left((\alpha-1)\,\mathcal{D}_\alpha\left(\mathbb{P}_{X_{1:k}} \parallel \mathbb{P}_{Y_{1:k}}\right)\right) &= \int_{\mathcal{X}_1 \times \cdots \times \mathcal{X}_k} \left[\frac{\mathbb{P}_{X_{1:k}}(x_{1:k})}{\mathbb{P}_{Y_{1:k}}(x_{1:k})}\right]^\alpha d\mathbb{P}_{Y_{1:k}}(x_{1:k}) \\
&= \int_{\mathcal{X}_1 \times \cdots \times \mathcal{X}_k} \left[\prod_{i=1}^k \frac{\mathbb{P}_{X_i|X_{1:i-1}=x_{1:i-1}}(x_i)}{\mathbb{P}_{Y_i|Y_{1:i-1}=x_{1:i-1}}(x_i)}\right]^\alpha \prod_{i=1}^k d\mathbb{P}_{Y_i|Y_{1:i-1}=x_{1:i-1}}(x_i) \\
&\leqslant \prod_{i=1}^k \sup_{x_1 \in \mathcal{X}_1,\ldots,x_{i-1} \in \mathcal{X}_{i-1}} \int_{\mathcal{X}_i} \left[\frac{\mathbb{P}_{X_i|X_{1:i-1}=x_{1:i-1}}(x_i)}{\mathbb{P}_{Y_i|Y_{1:i-1}=x_{1:i-1}}(x_i)}\right]^\alpha d\mathbb{P}_{Y_i|Y_{1:i-1}=x_{1:i-1}}(x_i) \\
&= \exp\left((\alpha-1)\sum_{i=1}^k \sup_{x_1 \in \mathcal{X}_1,\ldots,x_{i-1} \in \mathcal{X}_{i-1}} \mathcal{D}_\alpha\left(\mathbb{P}_{X_i|X_{1:i-1}=x_{1:i-1}} \parallel \mathbb{P}_{Y_i|Y_{1:i-1}=x_{1:i-1}}\right)\right).
\end{aligned}
$$

The remaining case of $\alpha = 1$ follows by continuity (or by the Chain Rule for KL divergence).

## A.2  Proof of Lemma 2.11

Observe that the mixture distribution $(1-q)\mathcal{N}(0,\sigma^2 I_d) + q\left(\mathcal{N}(0,\sigma^2 I_d) * \mu\right)$ can be further decomposed as the mixture distribution

$$
(1-q)\mathcal{N}(0,\sigma^2 I_d) + q\left(\mathcal{N}(0,\sigma^2 I_d) * \mu\right) = \int \left[(1-q)\mathcal{N}(0,\sigma^2) + q\mathcal{N}(z,\sigma^2)\right] d\mu(z),
$$

where this integral is with respect to the weak topology of measures. Now argue as follows:

$$\sup_{\mu \in \mathcal{P}(R\mathbb{B}_d)} \mathcal{D}_\alpha \left( \mathcal{N}(0, \sigma^2 I_d) \, \big\| \, (1-q)\mathcal{N}(0, \sigma^2 I_d) + q\left(\mathcal{N}(0, \sigma^2 I_d) * \mu\right) \right)$$

$$= \sup_{\mu \in \mathcal{P}(R\mathbb{B}_d)} \mathcal{D}_\alpha \left( \mathcal{N}(0, \sigma^2 I_d) \, \big\| \, \int \left[ (1-q)\mathcal{N}(0, \sigma^2) + q\mathcal{N}(z, \sigma^2) \right] d\mu(z) \right)$$

$$\leqslant \sup_{z \in \mathbb{R}^d \, : \, \|z\| \leqslant R} \mathcal{D}_\alpha \left( \mathcal{N}(0, \sigma^2 I_d) \, \big\| \, (1-q)\mathcal{N}(0, \sigma^2 I_d) + q\mathcal{N}(z, \sigma^2 I_d) \right)$$

$$= \sup_{r \in [0, R]} \mathcal{D}_\alpha \left( \mathcal{N}(0, \sigma^2) \, \big\| \, (1-q)\mathcal{N}(0, \sigma^2) + q\mathcal{N}(r, \sigma^2) \right)$$

$$= \mathcal{D}_\alpha \left( \mathcal{N}(0, \sigma^2) \, \big\| \, (1-q)\mathcal{N}(0, \sigma^2) + q\mathcal{N}(R, \sigma^2) \right)$$

$$= \mathcal{D}_\alpha \left( \mathcal{N}(0, (\sigma/R)^2) \, \big\| \, (1-q)\mathcal{N}(0, (\sigma/R)^2) + q\mathcal{N}(1, (\sigma/R)^2) \right)$$

$$= S_\alpha(q, \sigma/R).$$

Above, the second step is by the quasi-convexity of the Rényi divergence (Lemma 2.6), the third step is by the rotation-invariance of the Gaussian distribution, and the fifth step is by a change of variables. Finally, observe that all steps in the above display hold with equality if $\mu$ is taken to be the Dirac distribution at a point of norm $R$.

### A.3 Proof of Lemma 2.16

Let $\nu$ be a probability distribution that certifies $\mathcal{D}_\alpha^{(z)}(\mu \| \mu')$; that is, $\nu$ satisfies $\mathcal{D}_\alpha^{(z)}(\mu \| \mu') = \mathcal{D}_\alpha(\nu \| \mu')$ and $W_\infty(\nu, \mu) \leqslant z$.

We claim that

$$W_\infty \left( \phi'_\# \nu, \phi_\# \mu \right) \leqslant s + z. \tag{A.1}$$

To establish this, first use the triangle inequality for the Wasserstein metric $W_\infty$ to bound

$$W_\infty \left( \phi'_\# \nu, \phi_\# \mu \right) \leqslant W_\infty \left( \phi'_\# \nu, \phi_\# \nu \right) + W_\infty \left( \phi_\# \nu, \phi_\# \mu \right).$$

For the first term, pushforward $\nu$ under the promised coupling of $(\phi, \phi')$ in order to form a feasible coupling for the optimal transport distance that certifies $W_\infty(\phi'_\# \nu, \phi_\# \nu) \leqslant s$. For the second term, use the fact that $\phi$ is almost surely a contraction to bound $W_\infty(\phi_\# \nu, \phi_\# \mu) \leqslant W_\infty(\nu, \mu) \leqslant z$.

Now that we have established (A.1), it follows that $\phi'_\# \nu$ is a feasible candidate for the optimization problem $\mathcal{D}_\alpha^{(z+s)}(\phi_\# \mu \| \phi'_\# \mu')$. That is,

$$\mathcal{D}_\alpha^{(z+s)}(\phi_\# \mu \| \phi'_\# \mu') \leqslant \mathcal{D}_\alpha(\phi'_\# \nu \| \phi'_\# \mu').$$

By the quasi-convexity of the Rényi divergence (Lemma 2.6), the post-processing inequality for the Rényi divergence (Lemma 2.7), and then the construction of $\nu$,

$$\mathcal{D}_\alpha \left( \phi'_\# \nu \, \| \, \phi'_\# \mu' \right) \leqslant \sup_{h \in \text{supp}(\phi')} \mathcal{D}_\alpha \left( h_\# \nu \, \| \, h_\# \mu' \right) \leqslant \mathcal{D}_\alpha(\nu \| \mu') = \mathcal{D}_\alpha^{(z)}(\mu \| \mu').$$

Combining the last two displays completes the proof.

### A.4 Proof of Theorem 4.1

Assume $T \geqslant \bar{T}$ because once the theorem is proven in this setting, then the remaining setting $T \leqslant \bar{T}$ follows by the strong composition rule for RDP (Lemma 2.8) by equivalently re-interpreting the algorithm NOISY-SGD run once with many iterations as running multiple instantiations of NOISY-SGD, each with few iterations. Consider the choice of constants in Footnote 11; then for $T \geqslant \bar{T}$, the quantity in (4.1) is greater than 0.005. So it suffices to show that NOISY-SGD does not satisfy $(100, 0.005)$-RDP. By the conversion from RDP to DP (Lemma 2.5), plus the calculation $0.005 + (\log 100)/99 < 0.1$, it therefore suffices to show that NOISY-SGD does not satisfy $(0.1, 0.01)$-DP.

Consider the construction of datasets $\mathcal{X}, \mathcal{X}'$, functions $f$, and processes $\omega_t, \omega'_t, \omega''_t$ in §4. Then, in order to show that NOISY-SGD does not satisfy $(0.1, 0, 01)$-DP, it suffices to show that $\mathbb{P}\left[\omega'_T \geqslant 0\right] \geqslant e^{0.1}\mathbb{P}[\omega_T \geqslant 0] + 0.01$. We simplify both sides: for the left hand side, note that $\omega'_t$ stochastically dominates $\omega''_t$ for all $t$; and on the right hand side, note that $\mathbb{P}[\omega_T \geqslant 0] = 1/2$ by symmetry of the process $\{\omega_t\}$ around 0. Therefore it suffices to prove that

$$\mathbb{P}\left[\omega''_T \geqslant 0\right] \overset{?}{\geqslant} \frac{1}{2} e^{0.1} + 0.01. \tag{A.2}$$

To this end, we make two observations that collectively formalize the Gaussian approximation described in the proof sketch in §4. Below, let $Y = \sum_{t=T-\bar{T}}^{T-1} Y_t$ denote the total bias in the process $\omega''_t$, and let $E$ denote the event that both

(i) Concentration of bias in the process $\omega_t''$: it holds that $Y \in [1 \pm \Delta] \cdot \mathbb{E}Y$, for $\Delta = 0.15$.

(ii) No projections in the process $\omega_t''$: it holds that $\max_{t \in \{T - \bar{T}, \ldots, T\}} \omega_t'' < D/2$.

**Observation A.1** ($E$ occurs with large probability). $\mathbb{P}[E] \geqslant 0.9$.

*Proof.* For item (i) of $E$, note that $B := bY/(\eta L) = \sum_{t=T-\bar{T}}^{T-1} \mathbb{1}_{Y_t \neq 0}$ is a binomial random variable with $\bar{T}$ trials, each of which has probability $b/n$, so $B$ has expectation $\mathbb{E}B = b\bar{T}/n = 0.75bD/(L\eta)$. Thus, by a standard Chernoff bound (see, e.g., [33, Corollary 4.6]), the probability that (i) does not hold is at most

$$\mathbb{P}\Big[\text{item (i) fails}\Big] = \mathbb{P}\Big[B \notin [1 \pm \Delta] \cdot \mathbb{E}B\Big] \leqslant 2\exp\left(-\frac{\Delta^2 \cdot \mathbb{E}B}{3}\right) \leqslant 2\exp\left(-\frac{0.15^2 \cdot 0.75 \cdot 1000}{3}\right) \leqslant 0.01\,.$$

Next, we show that conditional on (i), item (ii) fails with low probability. To this end, note that (ii) is equivalent to the event that $\sum_{s=T-\bar{T}}^{t-1}(Y_s + Z_s) < D$ for all $t$. Thus because $\sum_{s=T-\bar{T}}^{t-1} Y_s \leqslant Y \leqslant (1+\Delta)\mathbb{E}Y = (1+\Delta)0.75D \leqslant 0.9D$ conditional on event (i), we have that

$$\mathbb{P}\Big[\text{item (ii) fails} \,\Big|\, \text{item (i) holds}\Big] \leqslant \mathbb{P}\left[\max_{t \in \{T-\bar{T},\ldots,T\}} \sum_{s=T-\bar{T}}^{t-1} Z_s \geqslant 0.1D\right].$$

Now the latter expression has a simple interpretation: it is the probability that a random walk of length $\bar{T}$ with i.i.d. $\mathcal{N}(0, \eta^2\sigma^2)$ increments never surpasses $0.1D$. By a standard concentration inequality on the hitting time of a random walk (e.g., see the application of Doob's Submartingale Inequality on [45, Page 139]), this probability is at most

$$\cdots \leqslant \exp\left(-\frac{(0.1D)^2}{2\bar{T}\eta^2\sigma^2}\right) \leqslant \exp(-5) \leqslant 0.01\,,$$

Putting the above bounds together, we conclude the desired claim:

$$\mathbb{P}\Big[E\Big] = \mathbb{P}\Big[\text{item (i) holds}\Big] \cdot \mathbb{P}\Big[\text{item (ii) holds} \,\Big|\, \text{item (i) holds}\Big] \geqslant (1 - 0.01)^2 \geqslant 0.9\,.$$

$\square$

**Observation A.2** (Gaussian approximation of $\omega_T''$ conditional on $E$). *Denote $Z := \sum_{t=T-\bar{T}}^{T-1} Z_t$. Conditional on the event $E$, it holds that $\omega_T'' \geqslant Z + 10/D$.*

*Proof.* By item (ii) of the event $E$, no projections occur in the process $\{\omega_t''\}$, thus $\omega_T'' = -D/2 + \sum_{t=T-\bar{T}}^{T-1}(Y_t + Z_t) = -D/2 + Y + Z$. By item (i), the bias $Y \geqslant (1-\Delta)\mathbb{E}Y = (1-0.15)0.75D \geqslant 0.6D$. $\square$

Next, we show how to combine these two observations in order to approximate $\omega_T''$ by a Gaussian, and from this conclude a lower bound on the probability that $\omega_T''$ is positive. We argue that

$$\begin{aligned}
\mathbb{P}[\omega_T'' \geqslant 0] &\geqslant \mathbb{P}[\omega_T'' \geqslant 0, E] \\
&\geqslant \mathbb{P}[Z + D/10 \geqslant 0, E] \\
&= \mathbb{P}[Z + D/10 \geqslant 0] - \mathbb{P}[Z + D/10 \geqslant 0, E^C] \\
&\geqslant \mathbb{P}[Z + D/10 \geqslant 0] - 0.1,
\end{aligned}$$

where above the second step is by Observation A.2, and the final step is by Observation A.1. Since $Z$ is a centered Gaussian with variance $\bar{T}\eta^2\sigma^2 \leqslant c_\sigma D^2 = 0.001D^2$, we have

$$\mathbb{P}[Z + D/10 \geqslant 0] \geqslant \mathbb{P}\Big[\mathcal{N}(0.1D, 0.001D^2) \geqslant 0\Big] = \mathbb{P}\Big[\mathcal{N}(0,1) \geqslant -0.1/\sqrt{0.001}\Big] \approx 0.999993.$$

By combining the above two displays, we conclude that

$$\mathbb{P}\Big[w_T'' \geqslant 0\Big] \geqslant 0.99993 - 0.1 \geqslant \frac{1}{2}\,e^{0.1} + 0.01.$$

This establishes the desired DP lower bound (A.2), and therefore proves the theorem.