# OpenReview forum: "Privacy of Noisy Stochastic Gradient Descent: More Iterations without More Privacy Loss"
_NeurIPS.cc/2022/Conference — NeurIPS 2022 Accept_

### Official Review · Reviewer_gSaZ · 2022-07-06

**Rating:** 7
**Confidence:** 3
**Soundness:** 3 good
**Presentation:** 4 excellent
**Contribution:** 3 good

**Summary:**

The authors show that Noisy SGD achieves a privacy loss that does not grow indefinitely in the setting of convex optimization with smooth Lipchitz functions over a bounded set. The proof technique relies on privacy amplification by sampling and on amplification by iteration. Additionally, the authors also show a lower bound showing that the upper bound is tight up to a constant.

**Questions:**

No questions

**Strengths And Weaknesses:**

Strenghts: Paper is well written, easy to read, provides enough background on the problem and a good literature review. The result is also pretty cool since it shows that previous upper bounds for the privacy loss, where privacy scales indefinitely, exhibit the wrong qualitative behavior.
It looks like the only similar results (a bounded privacy loss) only hold in the case of smooth and strongly convex setting, so being able to relax strong convexity is significant.
Weaknesses: I don’t have much to say here, I think it is a good paper.

---

> ### Author Response · Authors · 2022-08-02
> **Response to Reviewer gSaZ**
>
> Thank you very much for the overwhelmingly positive review. We are glad that you think this is a good paper, that the result is cool, and that you found the paper well-written and easy to read.

---

### Official Review · Reviewer_qRoi · 2022-07-10

**Rating:** 6
**Confidence:** 4
**Soundness:** 3 good
**Presentation:** 2 fair
**Contribution:** 3 good

**Summary:**

In this paper, authors show that running DP-SGD for a convex function on a bounded domain converges to a bounded privacy cost opposed to the classical composition results that grow to infinity as the number of iterations grow. Similar to Chourasia et al. 2021, authors assume that the internal state (i.e. the intermediate parameter values) are kept private and only the final iterate is released. Authors build a novel analysis, where they split the the parameter trace in two halves and analyse the privacy cost of these halves separately. This involves finding a new improved privacy amplification by iteration (PABI) result. Authors also present a lower bound result for the privacy guarantee.

**Questions:**

I don't have much to ask, since I found the proofs quite easy to follow. Couple of very minor things that caught my eye:
- Lines 303-304: "Notice also that this bias term is only realized with probability 1 − b/n because the probability that i∗ is in a random size-b subset of [n] is b/n.", maybe I miss something, but I thought the noise is realized w.p. b/n
- The bound in beginning of page 14 of supplement: I do understand that if you set $R=\tilde{T}$, the summands inside the $\min$ get equivalent and with larger $R > \tilde{T}$ the latter term gets smaller and and former grows. However, I'm not quite sure, how do you form the last inequality. It almost seems like you only have the second term of the summand in the final expression, and the first has vanished somewhere.
- I'm curious, could one use some adaptive step-size schemes like Adam with this final iterate release analysis? Or would the Adam cause some sort of side-channel privacy leak through the last gradients it uses in adapting the step-size?

**Limitations:**

There is not much discussion of the limitations, probably due to lack of space. I guess the authors could further discuss the limitations that the assumptions make. For example the assumption of bounded domain.

**Strengths And Weaknesses:**

The main result of the paper (the upper bound) is a significant finding. Although, Chourasia et al. 2021 already showed this type of behaviour of the privacy cost for an iterative algorithm, this paper presents following significant updates to the result of Chourasia el al; extending the result for subsampled GD (preferable for computational reasons), and replacing the strong convexity assumption with just convexity.

Authors have done significant amount of original work to solve the question at hand. The proof of main theorem (Thm. 3.1) comprises of many novel techniques and results, such as the new PABI result. I think the main limitation of the result is the assumption of limited search domain. However it is not a too strong of an assumption, and I think there are several real world problems that do have a priori known limited optimization domain.

To me, the most apparent weakness of the work is the presentation. The supplement included in the submission seemed like a (almost) perfectly good journal version (~19 pages) of the article, and was really pleasant to read. However, the cut down version is way too crowded currently. I do appreciate authors efforts to have the most crucial bits in the main paper, but I think the paper needs even more polishing and moving less important stuff to the appendix to make it to the NeurIPS format. And I'm afraid that it is going to be really difficult given the amount of material this paper contains.This lack of space also gives the paper a really abrupt ending with no discussion or conclusions.

---

> ### Author Response · Authors · 2022-08-02
> **Response to Reviewer qRoi**
>
> We are very grateful to the reviewer for the kind words that our paper is a significant finding, has done a significant amount of original work, that the proofs are quite easy to follow, and that the supplement seemed like a (almost) perfectly good journal version.
>
>
>
> ---> The reviewer brings up two concerns, one about assumptions and one about presentation.
>
> 1. Regarding assumptions: We kindly point the reviewer to the detailed discussion on page 4 and elaborate here.
>
> 1a. Main assumptions: Briefly, our three assumptions (boundedness, smoothness, and convexity) are in fact necessary and sufficient for the privacy of Noisy-SGD to converge (our main result). Indeed, on one hand, for any strict subset of these three assumptions, there are counterexamples where the privacy breaks down completely as T -> infty. And on the other hand, our main result establishes that these three assumptions are sufficient. Therefore our paper leads to a full understanding of the problem and its reliance on assumptions. With the extra page afforded by the camera-ready, we will add more discussion to clarify this.
>
> 1b. Diameter: We agree with the reviewer that “it is not too strong of an assumption, and … there are several real world problems that do have a priori known limited optimization domain.” Moreover, we would like to point out two things. First, every optimization/utility guarantee for (non strongly) convex losses also has a similar dependence on some diameter bound; in fact this is inevitable simply due to the difference between initialization and optimum. Second, one can solve an unconstrained problem by solving constrained problems with norm bounds on the optimum, paying only a small logarithmic overhead on the number of solves and a small constant overhead in privacy using known techniques (https://arxiv.org/abs/1811.07971). Therefore, this diameter bound is a mild assumption in practice and also is in common with all existing theoretical bounds for utility.
>
> 2. Regarding presentation: We are very grateful to the reviewer for saying that the supplement “seemed like a (almost) perfectly good journal version, and was really pleasant to read”. The camera-ready version of the paper will give us an extra page, which will let us add more discussion and make the presentation less crowded. We will also polish the paper as the reviewer suggests in the camera-ready, by moving some more content to the Supplementary. This will significantly improve the exposition.
>
>
> ---> Response to minor questions:
>
> 1. Correct, the Gaussian noise is always there. By “bias is realized”, we mean that this noise is possibly non-centered with this probability. We will clarify this wording in the camera-ready.
>
> 2. The term is safely dropped since it is lower order. Indeed, 1/n^2 <= R/n^2 since R is at least 1. We will clarify this wording in the camera-ready.
>
> 3. Fantastic question. You are exactly right that the main technical obstacle there is how to control the privacy leakage from how past iterates affect the adaptivity in later iterates. This appears to preclude using our analysis techniques, at least in their current form. It would be very exciting for both theory and practice if one could find some way of extending our results to adaptive methods, and we will highlight this as an interesting open question for future work in the camera-ready version.

---

### Official Review · Reviewer_iTwN · 2022-07-10

**Rating:** 7
**Confidence:** 3
**Soundness:** 3 good
**Presentation:** 3 good
**Contribution:** 3 good

**Summary:**

This work proposes an asymptotic privacy bound analysis when $T$ is large in Noisy SGD (SGLD) for convex functions. The results show when the number of iterations $T$ increases at the start, the privacy decreases at a linear rate. After a burn-in period, the privacy is bounded by a constant, and running SGD longer leaks no further privacy.  Some techniques based on PABI are also proposed in the analysis. The work also shows the tightness with a lower bound construction based on the trace analysis of random walk, which makes the proposed magnitude more convincing.

**Questions:**

Some questions:
1. Can the proposed framework be extended to the bounded non-convex function with a regularizer, like [Li et al, 2019] for SGLD (Also mentioned in [Chourasia et al. 2021]).
2. How can the proposed bounds be used/ help the cost minimization and privacy mechanism design?
3. How a $D$ determined for the real cases? Is $D$ related to the dimension and the convergence of the convex problem? Will the convergence influence the privacy level?


Some suggestions:
1. Some simple examples/numerical studies can be provided to show the effectiveness of the proposed framework more intuitively.


**Ethics Review Area:**

["I don’t know"]

**Limitations:**

The assumptions have been discussed in detail, showing the limitations of the work.

**Strengths And Weaknesses:**

Pros:
1. The paper discusses the core problem in the Noisy-SGD analysis and provides a new asymptotic viewpoint for the privacy bound. Some new techniques are also proposed for the problem
2. The paper is well-organized and some important points and proof sketches are shown in the main context with the detailed proof in the Supplement.
3. Good and detailed sketches for the existing Noisy-SGD analysis.

Cons:
1. The paper only discusses the function for convex cases, which simplifies the difficulties for analysis. More works might focus on the analysis in non-convex cases (like deep networks).
2. The paper lacks some numerical studies to give more insights and verify the performance of the proposed framework.

---

> ### Author Response · Authors · 2022-08-02
> **Response to Reviewer iTwN**
>
> We are very grateful to the reviewer for the kind words about our result, the presentation of the paper, the discussion of assumptions, and the technical sketches.
>
> ----> Response to the two main concerns:
>
> 1. Regarding non-convexity: The purpose of this paper is to resolve the privacy leakage for convex losses since the problem was open even in this foundational setting. We agree that extensions to non-convex losses would be very interesting. Unfortunately, convexity is necessary for the privacy to converge as T -> infinity, and in fact we know of simple non-convex counterexamples. Any such extension would therefore have to make additional structural assumptions on the non-convexity of the function (and possibly also change the Noisy-SGD algorithm), although it is unclear how exactly this would even look. Moreover, this appears to require significant new machinery as our techniques are the only known way to solve the convex problem, and they break down in the non-convex setting (see the detailed discussion about this in L136-145). In the camera-ready, we will add these comments to the discussion of convexity in L136-145.
>
> 2. Regarding numerics: Our hope is that because our main result lets us run Noisy-SGD forever without leaking more privacy, this may enable training higher accuracy models with the same privacy budget in many settings. We agree that it would be very interesting to have a thorough empirical investigation of when this does and does not help, and what properties of the optimization problems and datasets influence this. A detailed investigation of these practical considerations is unfortunately outside the scope of this theoretical work. We remark in passing that in Appendix B, we provide tight numerical versions of our main result to make it very easy for others to use and implement our improved privacy bounds.
>
> ----> Response to questions:
>
> 1. Interesting question. Our techniques are currently the only way of proving convergent privacy bounds for convex losses, and this breaks down without the contractivity of a gradient descent step which is implied by convexity. See (1) above for details.
>
> 2. Also a good question. Our techniques readily generalize to any iterative algorithm which interleaves contractive steps and noise convolutions. Such algorithms are common in differentially private optimization, and it would be interesting to apply them to variants of Noisy-SGD.
>
> 3. In many optimization problems, the solution set is naturally constrained either from the problem formulation or application. We also mention in passing that one can solve an unconstrained problem by solving constrained problems with norm bounds on the optimum, paying only a small logarithmic overhead on the number of solves. The privacy overhead of doing so can be bounded by a constant using known techniques (https://arxiv.org/abs/1811.07971). Therefore this is a mild assumption.
>
> We will add discussion of this to the camera-ready version.

---

### Official Review · Reviewer_v3zG · 2022-07-12

**Rating:** 6
**Confidence:** 4
**Soundness:** 3 good
**Presentation:** 3 good
**Contribution:** 3 good

**Summary:**

This paper studies the privacy loss of noisy projected stochastic gradient descent. More specifically, the authors show that after some iterations, noisy projected stochastic gradient will not cause further privacy leakage if we only release the last iterate of the algorithm. This result looks very interesting and promising for private convex optimization.

**Questions:**

I think the result of the current paper is very interesting, and my main concerns are as follows:
1. How to make use of the new analysis to better understand the utility of noisy SGD.
2. The new analysis seems to be limited to noisy projected SGD. However, in practice, people often clip the stochastic gradient and then add noisy, and how can this analysis help us to understand the privacy loss of the widely used clipped based method? In other words, whether practitioners can benefit from your analysis in practice to apply noisy project SGD?

**Limitations:**

Yes

**Strengths And Weaknesses:**

Strengths:
1. The authors provide a new privacy analysis for noisy projected stochastic gradient descent when optimizing a convex, Lipschitz, and smooth objective with a bounded parameter space.
2. The new privacy loss upper bound shows that noisy projected stochastic gradient descent will not cause further privacy loss after a certain number of iterations.

Weaknesses:
1. It is unclear how this result can be used to study the utility guarantee of noisy SGD. According to Bassily et al, 2019, the original private analysis can achieve the optimal utility guarantee for noisy SGD, and thus it is unclear how we can make use of the new privacy analysis.
2. One drawback of the current analysis is that we can only release the last iterate.

---

> ### Author Response · Authors · 2022-08-02
> **Response to Reviewer v3zG**
>
> We are very grateful to the reviewer for the kind words that our result is very interesting and promising for private optimization.
>
> ----> Response to concerns:
>
> 1. The reviewer is correct that theoretically, minimax utility bounds for DP ERM and DP SCO are obtained already with existing analyses, i.e. where one does not need to run for too long. For example, for (smooth convex) DP SCO (and for non-private SCO for that matter), one epoch of SGD is enough to achieve the minimax rate (https://arxiv.org/abs/2005.04763). However, this is a big difference between theory and practice because in practice Noisy-SGD benefits from running longer to get more accurate training. In fact, this divergence is even true and well-documented for non-private SGD as well, where one epoch is minimax-optimal in theory, but in practice more epochs help. Said simply, this is because typical problems are not worst-case problems (i.e., minimax-optimality theoretical bounds are typically not representative of practice). For these practical settings, in order to run Noisy-SGD longer, it is essential to have privacy bounds which do not increase. Our paper resolves this.
>
> 2. Publishing the whole path of SGD over multiple epochs is provably not private. One may hope that an average iterate may have better privacy properties, but as shown in (https://arxiv.org/abs/2005.04763), there are fundamental difficulties with trying to prove privacy even of the average iterate: indeed, the average iterate is provably not as private for Cyclic Noisy-SGD.
>
> ---> Response to questions:
>
> 1. As discussed in (1) above, the utility-privacy tradeoff is fully understood in the minimax sense. Our new and optimal privacy analysis may help get better utility-privacy tradeoffs in practical settings that are not captured by worst-case analyses. Specifically, privacy-utility tradeoffs are obtained from combining two bounds which are proved separately: (i) privacy of the algorithm as a function of the # of iterations, (ii) utility of the algorithm as a function of the # of iterations. The purpose of this paper is to completely resolve (i); this result can then be combined with any bound on (ii).
>
> 2. Noisy SGD with clipping is used in practice for non-convex as well as convex problems. In the common case of GLMs, the clipped gradients can be viewed as gradients of a different convex loss (https://arxiv.org/abs/2006.06783), in which case our results can be directly applied. In general, clipped gradients do not correspond to gradients of a convex loss, in which case our results (as well as all other works in the literature that aim at proving convergent privacy bounds) do not apply.
>
> We will add discussion of this to the camera-ready.

---

### Meta-Review · Area_Chair_Dp6D · 2022-08-22

**Recommendation:** Accept
**Confidence:** Certain

**Metareview:**

The paper solves a longstanding open problem of showing bounded privacy loss for releasing the last iterate of noisy SGD for convex problems This improves upon previous work by going from GD to SGD and strongly convex to convex. All reviewers agree this is a strong paper and should be accepted.

**Award:**

No

---

### Decision · Program_Chairs · 2022-09-14

Accept